# Monitoring Ground Surface Deformation of Ice-Wedge Polygon Areas in Saskylakh, NW Yakutia, Using Interferometric Synthetic Aperture Radar (InSAR) and Google Earth Engine (GEE)

Wenhui Wang [1], Huijun Jin [1,2,3,*], Ze Zhang [1,†], Mikhail N. Zhelezniak [4,†], Valentin V. Spektor [4], Raul-David Șerban [2,5], Anyuan Li [3], Vladimir Tumskoy [4], Xiaoying Jin [1], Suiqiao Yang [1], Shengrong Zhang [1], Xiaoying Li [1], Mihaela Șerban [6], Qingbai Wu [2] and Yanan Wen [7]

1  School of Civil Engineering, Permafrost Institute, and School of Forestry, Northeast Forestry University, Harbin 150040, China
2  State Key Laboratory of Frozen Soils Engineering, Northwest Institute of Eco-Environment and Resources, Chinese Academy of Sciences, Lanzhou 730000, China
3  School of Civil Engineering, Shaoxing University, Shaoxing 312000, China
4  Melnikov Permafrost Institute, Russian Academy of Sciences, 677010 Yakutsk, Russia
5  Institute for Alpine Environment, Eurac Research, 39100 Bolzano, Italy
6  Applied Geomorphology and Interdisciplinary Research Centre, Department of Geography, West University of Timisoara, 300223 Timișoara, Romania
7  College of Land Science and Technology, China Agricultural University, Beijing 100083, China
*  Correspondence: hjjin@nefu.edu.cn; Tel.: +86-188-9318-5876
†  These authors contributed equally to this work.

**Abstract:** As one of the best indicators of the periglacial environment, ice-wedge polygons (IWPs) are important for arctic landscapes, hydrology, engineering, and ecosystems. Thus, a better understanding of the spatiotemporal dynamics and evolution of IWPs is key to evaluating the hydrothermal state and carbon budgets of the arctic permafrost environment. In this paper, the dynamics of ground surface deformation (GSD) in IWP zones (2018–2019) and their influencing factors over the last 20 years in Saskylakh, northwestern Yakutia, Russia were investigated using the Interferometric Synthetic Aperture Radar (InSAR) and Google Earth Engine (GEE). The results show an annual ground surface deformation rate (AGSDR) in Saskylakh at −49.73 to 45.97 mm/a during the period from 1 June 2018 to 3 May 2019. All the selected GSD regions indicate that the relationship between GSD and land surface temperature (LST) is positive (upheaving) for regions with larger AGSDR, and negative (subsidence) for regions with lower AGSDR. The most drastic deformation was observed at the Aeroport regions with GSDs rates of −37.06 mm/a at tower and 35.45 mm/a at runway. The GSDs are negatively correlated with the LST of most low-centered polygons (LCPs) and high-centered polygons (HCPs). Specifically, the higher the vegetation cover, the higher the LST and the thicker the active layer. An evident permafrost degradation has been observed in Saskylakh as reflected in higher ground temperatures, lusher vegetation, greater active layer thickness, and fluctuant numbers and areal extents of thermokarst lakes and ponds.

**Keywords:** climate change; ice-wedge polygon (IWP); ground surface deformation (GSD); interferometric synthetic aperture radar (InSAR); Google Earth Engine (GEE); Russian Arctic

## 1. Introduction

Ice-wedge polygons (IWPs) are important components of the periglacial landscapes, which plays important roles in the geomorphologic, stratigraphical, hydrological, and paleoenvironmental dynamics at present and in the past, such as those in the Late Pleistocene [1]. IWPs are characterized by their polygonal distributive patterns of surficial

deposits [2]. In connection with ongoing climate warming, it is presumed that permafrost will be transformed from a long-term carbon sink to a major carbon source due to the subsequent carbon release from these thawing organic/ice-rich sediments [3,4]. IWPs are generally found in flat watersheds, the bottoms of drained lakes, river terraces, and floodplains in the tundra with high ice content and with huge carbon reserves; with climate warming, ground ice melts and the ground settles, emitting the long-stored organic carbon [5–7]. They extensively occur in the arctic coastal plains of Alaska, the middle and low Arctic of Canada, and northern Russia (including northeastern Siberia) [2,7]. Since IWPs are one of the best and most reliable indicators of periglacial conditions, understanding the dynamics of IWPs is extremely important to evaluate the hydrothermal state of permafrost and carbon budgets [8]. Meanwhile, IWPs are one of the surface manifestations of yedoma, or ice complexes [9,10]. A better and more systematic research of IWP inventory will substantially contribute to the understanding of yedoma formation and decay processes and mechanisms, ice-wedge deposits, and landscapes.

The decaying of ice wedges in the Arctic has accelerated dramatically in the last three decades due to above-average summer temperatures [11–14]. Dynamic changes in IWPs have been studied by field and remote sensing methods from different perspectives by many scholars worldwide. Numerous studies have been conducted on the mapping of IWP through satellite remote sensing images, airborne light detection and ranging (LiDAR) of different resolutions, and machine learning methods [15–20]. Many field experiments and numerical model simulations have been employed to study the dynamics of IWP, e.g., the influences of IWP microtopography on near-surface ground temperatures [21], numerical models of IWP drainage [22], miniature accelerometers-based detection of ice-wedge cracking [23], field observations of syngenetic IWPs [24], automated multi-sensor-based ground displacements, hydrothermal conditions monitoring [8], and vulnerability and degradation of ice wedges [25]. The abovementioned studies have investigated dynamic changes in IWPs from different perspectives by remote sensing and/or ground-based methods. However, it is difficult to massively monitor the ground surface deformation (GSD) of IWPs because of the limitations of optical remote sensing for deformation measurement, painstaking workloads, and prohibitive costs of ground-based measurements. The GSD of IWP reflects the dynamics of the underlying permafrost in response to hydroclimatic changes and other disturbances [26]. Therefore, it is important to timely and properly extract the information on GSD of IWPs and the related environmental controllers or influencing factors.

Saskylakh is one of the key towns in Siberia where the man-fabricated surfaces (e.g., Saskylakh Aeroport) are in good contrast with different types of IWPs and thermokarst lakes and ponds. Therefore, the geocryological study in Saskylakh enables a comparative study of IWP and technogenic surfaces. In our study, Small Baseline Subset Interferometric Synthetic Aperture Radar (SBAS-InSAR) and Google Earth Engine (GEE) were used to monitor the GSD characteristics and the related environmental factors of IWP in Saskylakh, Anabarsky District, the Sakha (Yakutia) Republic, Russia. The objectives of this research paper are: (1) to obtain GSD characteristics of different types of IWPs by Sentinel-1 B-based SBAS-InSAR in the study area; (2) to study permafrost development by ground temperature, active layer thickness (ALT), and thermokarst lakes and ponds with Landsat 8 OLI/TIRS and Landsat 7 ETM+; and (3) to discuss changes in the permafrost environment and GSD of IWP areas by two transects and environmental factors.

## 2. Study Area and Sites in Saskylakh

The study area of Saskylakh (114.10°E, 71.95°N; ~17 m a. s. l.) is located on the eastern bank of the Anabar River in the northwestern Arctic Yakutia [27,28] (Figure 1a,b). The northern side of the study area is Saskylakh town, which is nearly 40 m higher than the study area of polygonal ground fields near the Saskylakh Aeroport. The eastern side is a terrace with sparse vegetation, which is nearly 20 m higher than the study area. The western side is bordered by the Anabar River; and on the southern side, there is a large

area of thermokarst lakes and ponds. The study area covers an areal extent of 12.84 km$^2$ of the IWP field in the continuous permafrost zone, with high ground ice content (>40% volumetric) [27] and ALT at about 1 m [28,29]. Based on the Saskylakh meteorological station (Registration No. RSM00021802) data [30,31] from 1944 to 2020, the local climate of Saskylakh is characterized by brief, mild summers and long, severely cold winters. The record maximum air temperature in summer is 35.6 °C (2 July 1979), and the record minimum air temperature in winter is −60.3 °C (12 December 2002). The multi-year average of annual precipitation was 115.54 mm, with a range from 78 mm (1950) to 517 mm (2018). The multi-year average of annual snow thickness was 45 cm, with a range from 7 cm (2012) to 110 cm (1973). In terms of monthly snowfall frequency, June and September had the least snowfall and the rest of the year had a similar frequency; in terms of monthly snowfall amount, June, November, and December had the least snowfall and the rest of the year had similar snowfall [32]. According to the soil classification system of the Food and Agriculture Organization (FAO) of the United Nation, soil types are fluvisol, regosol, and phaeozem. The stratigraphic age of rock types in Saskylakh is Permian, and there is no geological fault identified around the study area [33].

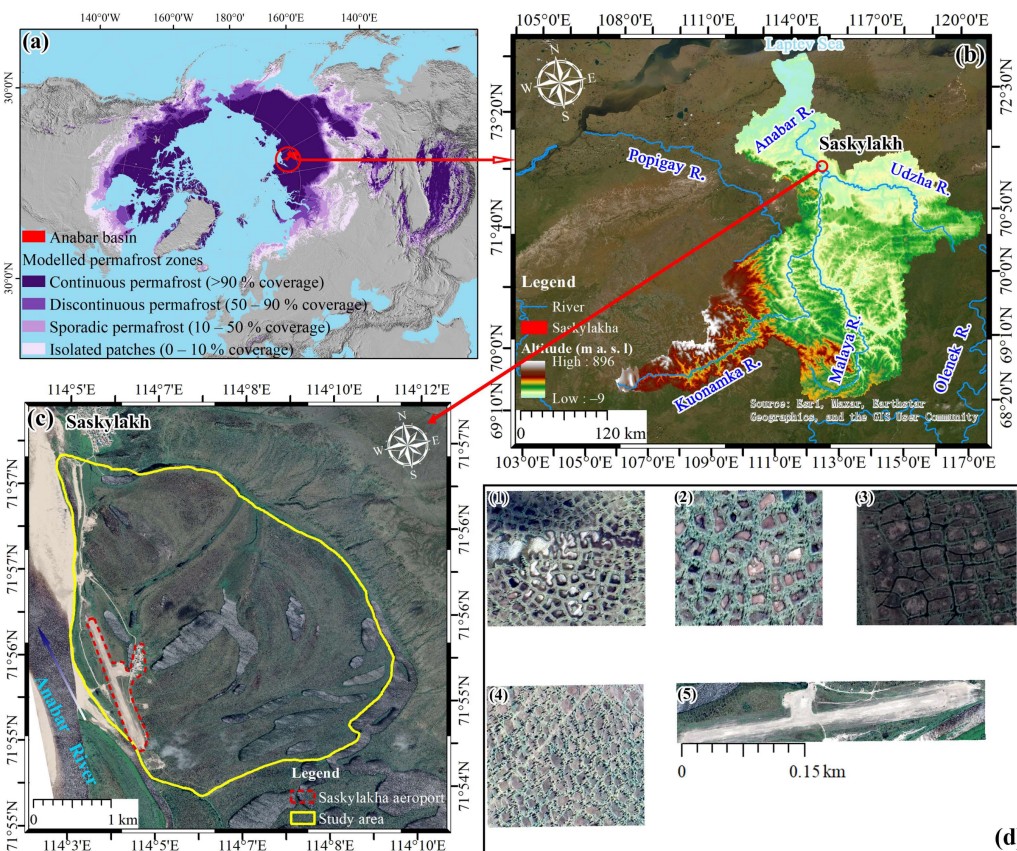

**Figure 1.** Location of the study area in Saskylakh, northwestern Yakutia, Russia. (**a**) Global permafrost distribution and study area location, Saskylakh, Anabarsky District, the Sakha Republic, Russia (URL: https://climate.esa.int/en/projects/permafrost/, accessed on 27 February 2023) cited from [28]; (**b**) Location of the study area in the Anabar basin (URL: https://www.hydrosheds.org/, accessed on 27 February 2023); (**c**) Study area image (the background image is a Google Earth image, accessed in August 2013); the yellow line is the study boundary; and (**d**) The representative examples of different IWPs. Insets (1) to (4) are the representative sample from Google earth images of LCP (1–2) and HCP (3–4) in the study area; and inset (5) is the representative sample of Google Earth images for the Saskylakh Aeroport.

According to the widely accepted classification method [34,35], Saskylakh has both low-centered polygons (LCPs) and high-centered polygons (HCPs). LCPs are generally found in flat areas with surficial peat accumulation and commences to grow as incipient polygons. However, with the gradual accumulation of peat and the displacement of material by the growing wedge ice, ridges usually develop parallel to the sides of the ice-wedge troughs. HCPs result from the lowering of the water level in the ice-wedge troughs of intermediate-centered polygons to leave domed areas above the troughs [36]. The study area is dominated by LCPs and HCPs, mosaicked by a relatively small proportion of thermokarst lakes and ponds, as well as lower-lying drainage channels with lusher vegetation. The Saskylakh Aeroport (61 m a. s. l., with a runway length of 1800 m) is the objective of the comparative study for their engineering or other anthropogenic disturbances.

## 3. Data and Methods

### 3.1. SAR Data

In this paper, we used Sentinel-1 B images released by the European Space Agency (ESA) [37,38]. To maintain good coherence, an one-year time (1 June 2018 to 3 May 2019) was selected for SBAS-InSAR. Sentinel-1 is an imaging radar mission for providing continuous all-weather, day-and-night imagery at C-band (5.54 cm in wavelength). The Sentinel-1 constellation provides high reliability, improved revisit time and geographical coverage, and rapid data dissemination to support operational applications in the priority areas of marine and land monitoring and emergency services. The Sentinel-1 B images were from the Alaska Satellite Facility (ASF, URL: https://vertex.daac.asf.alaska.edu/#, accessed on 27 February 2023) due to the high speed of downloading. Specific parameters are listed in Table 1 as follows.

**Table 1.** Sentinel-1 B parameters.

| Name | Parameters |
| --- | --- |
| Mode | Interferometric Wide (IW) swath |
| Product level | Level-1 Single Look Complex (SLC) |
| Revisit time | 12 days |
| Resolution (range × azimuth) | 5 × 20 m |
| Polarization | Vertical-vertical (VV) |
| Time period | 1 June 2018 to 3 May 2019 |

### 3.2. Optical Data

Because of the discontinuity and sometimes dissatisfactory quality of the single optical image, we adopted for an optical image combination strategy from different satellites. In this paper, the optical data include the Landsat 7 ETM+, Landsat 8 OLI/TIRS, and Moderate Resolution Imaging Spectroradiometer (MODIS). These data were atmospherically corrected. The cloud cover larger than 10% was filtered in the GEE. A filter of 10% cloud cover was used and only images of Landsat 7 ETM+ and 8 OLI/TIRS during summer met this requirement. They were used to extract the data for NDVI, MNDWI, and LST in the study area. DEM was downloaded from Japan Aerospace Exploration Agency (JSXA URL: https://www.eorc.jaxa.jp/ALOS/en/aw3d30/data/index.htm, accessed on 27 February 2023) (released in March 2017). These data were processed in GEE. Specific parameters are shown in Table 2.

### 3.3. Permafrost Data

Permafrost data used in this paper were provided by the ESA Permafrost Climate Change Initiative (CCI) 2.3 program (URL: https://climate.esa.int/en/odp/#/dashboard, accessed on 27 February 2023). The Permafrost CCI project is to establish Earth observation-based products for the permafrost essential climate variables (ECVs) spanning from 1997 to 2019. Since ground temperature and ALT cannot be directly observed from space-borne sensors, a variety of satellite and reanalysis data are combined in a ground thermal

model. The algorithm uses remotely sensed data sets of LST (MODIS LST/ESA LST CCI) and landcover (ESA Landcover CCI) to drive the CryoGrid CCI, a transient permafrost model [39]. The model computation yields ALT and ground temperatures at various depths (GST, T1, and T10, i.e., ground temperatures at the ground surface (5 cm in depth), 1, and 10 m in depth, respectively). Ground temperature serves as the basis for permafrost conditions and the horizontal resolution of the ALT and ground temperature products is 1000 m [40]. The permafrost data provided by permafrost CCI 2.3 were verified by 920 boreholes from the Global Terrestrial Network for Permafrost (GTN-P) and the Thermal State of Permafrost (TSP) project network and from China. The ALT provided by permafrost CCI 2.3 is highly consistent with the ALT provided by the Circumpolar Active Layer Monitoring (CALM) program [29].

**Table 2.** Optical and DEM data parameters used in this paper.

| Sensor | Spatial Resolution | Parameter | Bands | Unit | Range | Time | Value Type |
|---|---|---|---|---|---|---|---|
| Landsat 8 OLI/TIRS | 30 m | LST | B4, B5, B10 | °C | / | 2015~2020 | Year Average |
| | | NDVI | B4, B5 | / | −1~1 | | Year Average |
| | | MNDWI | B3, B6 | / | −1~1 | | Year Average |
| Landsat 7 ETM+ | | NDVI | B3, B4 | / | −1~1 | 2010~2015 | Year Average |
| | | MNDWI | B2, B5 | / | −1~1 | | Year Average |
| MODIS (MOD11A1) | 1000 m | LST | / | °C | / | 2000~2020 | Month Average |
| ALOS | 30 m | DEM | / | m | / | / | / |

### 3.4. SBAS-InSAR

For measuring GSD, there are various non-contact measurement methods, such as global positioning systems (GPSs), LiDAR, and optical remote sensing. Because of the high cost of GPS [41] and LiDAR, moderate quality imagery of optical remote sensing [42] in Saskylakh, northwestern Yakutia, the high-accuracy, wide-range, and weather-independent InSAR technology [41,43–45] was adopted for monitoring the GSD due to the limited accessibility of the study area in recent years due to the pandemic ensued issues and the high costs of ground-based GPS.

In combination with accurate SAR satellite imaging geometry and orbital ephemeris parameters, the SBAS-InSAR technique [46] takes advantage of the unique phase information of time-series SAR data. SBAS-InSAR obtains discrete targets that remain stable over a long period and can monitor GSD at the millimetric level, offering significant advantages for monitoring slow surface deformation. In this study, the SBAS-InSAR analysis was completed in SARscape 5.2.1.

The first step was to generate the connection map, where the super master image of the ascending tracks was automatically selected as the image of 28 May 2019, with the time baseline set at 90 days.

The second step was the interference process. Multi-looking was set to 4 × 1 in the range and the direction of the azimuth, minimum cost flow was used in phase unwrapping. After checking the quality of the interferograms, the pairs with too many systematic residual fringes and wrapped interferograms with very low coherence (no clear visible fringes) were removed.

The third step was re-flattening. In this step, we used the method of geometric control points (GCPs) to estimate and remove the remaining phase constants and phase ramps from the unwrapped phase stack. GCPs were selected based on the proximity of their deformation to the value of zero. This selection is confirmed by interpreting the unwrapped phase and evenly distributed GCPs points outside the phase jump region [47]. Consequently, 24 GCPs were applied outside refining orbital errors.

The fourth step was to invert the deformation using a linear inversion model to remove the residual topography and derive the preliminary displacement. Atmospheric Low Pass

filtering is related to the spatial characteristics of the atmosphere, for which the atmosphere is highly coherent in space. The atmospheric High Pass filtering is related to the temporal characteristics of the atmosphere, for which the atmosphere is low coherent in time. To remove the atmosphere residual error, the Atmosphere Low Pass Size was set to 800 m and the Atmosphere High Pass Size was set to 365 days. Eventually, the least squares solution obtained by the method of Singular Value Decomposition (SVD) was used for estimating the time series of the nonlinear GSD.

The fifth step was geocoding to invert the SBAS-InSAR information for GSD values. The dummy value was removed and the deformation results were interpolated. SBAS-InSAR-derived deformation was in the line-of-sight (LOS) direction. Then, the LOS direction was converted to the vertical surface deformation $d_V$ by the equation:

$$d_V = \frac{d_{LOS}}{cos\theta}$$

where $\theta$ is the angle of incidence of the satellite [45,48].

*3.5. Evaluation of Theoretical Precision*

Due to the lack of ground-based field Global Navigation Satellite System (GNSS) or leveling surveys to verify the SBAS-InSAR results, the atmospheric delay and phase unwrapping in the SBAS-InSAR processing process causes experimental errors, and these errors affect the accuracy of the SBAS-InSAR results to varying degrees. Thus, the phase variance formula was selected to verify the accuracy of the SBAS-InSAR results. The larger the phase variance, the lower the experiment accuracy and *vice versa*. The phase variance is expressed by the following formula:

$$\sigma_{\varphi}^2 = \frac{1 - \gamma^2}{2\gamma^2} \frac{\gamma}{4\pi}$$

where $\sigma_{\varphi}^2$ is the phase variance; and $\gamma$ is the interferometric coherence coefficient. Wherein, $\gamma$ can be obtained by the following formula:

$$\gamma = \frac{\left| \sum_{i=1}^{N} \sum_{j=1}^{M} \mu_1(i,j) \otimes \mu_2(i,j) \right|}{\sqrt{\sum_{i=1}^{N} \sum_{j=1}^{M} |\mu_1(i,j)|^2 \sum_{i=1}^{N} \sum_{j=1}^{M} |\mu_2(i,j)|^2}}$$

where $M$ and $N$ are the sizes of the data block to calculate the coherence; $i$ and $j$ are the row and column numbers in the data block; $\mu_1$ and $\mu_2$ are the complex values at the image coordinates $(i, j)$ in the main pair image data block; and the symbol $\otimes$ denotes the conjugate multiplication [49].

*3.6. Other Methods*

3.6.1. Normalized Difference Vegetation Index (NDVI)

Permafrost and vegetation interact hydrothermally, and NDVI comprehensively reflects the vegetation canopy, cover, phytomass, and other features [50]. This practice is widely adopted in the study of permafrost [51]. NDVI is defined as:

$$NDVI = \frac{(Nir - Red)}{(Nir + Red)}$$

where *Nir* and *Red* represent surface reflection averaged over a range of wavelengths in the near-infrared and red bands, respectively [50].

3.6.2. Modified Normalized Difference Water Index (MNDWI)

As proposed by Xu [52], MNDWI enhances the difference between the water body and its surrounding environment, and MNDWI is widely used in extracting water pa-

rameters [53]. MNDWI proved to be one of the indices with fewer misclassifications of thermokarst lakes on the northeastern Qinghai–Tibet Plateau compared with other remote sensing methods [54]. In this paper, MNDWI was used to distinguish the thermokarst ponds from other environmental settings and land use types. Thawing of ice-rich permafrost, or melting of massive ground ice, leads to GSD and enhanced soil erosion, resulting in depressions and later water-filled thermokarst ponds or lakes [54]. MNDWI is defined as:

$$MNDWI = \frac{(\text{GREEN} - \text{SWIR})}{(\text{GREEN} + \text{SWIR})}$$

where GREEN is the reflectivity of the green band, and SWIR is the shortwave infrared band. The area with *MNDWI* greater than zero is thus defined as thermokarst ponds or lakes.

### 3.6.3. Land Surface Temperature (LST)

As an important component of the Earth's energy balance and measured at the interface of the atmosphere and the surface materials, LST is closely related to the distribution of sensible and latent heat fluxes. LST is well-correlated with ground-based near-surface air temperature measured at heights of 1–3 m above the ground surface in the continuous permafrost zone [55,56]. This paper uses LST data with two different spatial and temporal resolutions: the Landsat 8 OLI/TIRS data with 30 m spatial resolution and a 16-day revisit period and the MODIS data with an 1000 m spatial resolution and an one-day revisit period. The LST was obtained using a statistical mono-window (SMW) algorithm in the GEE with the code provided by reference [57].

### 3.6.4. Selection of Ice-Wedge Polygon (IWP) Areas

GSD in permafrost areas is strongly related to variations in LST and ground ice content [58–60]. In this paper, the GSD characteristics were compared with MODIS-based LST data. By calculating the $\text{PCC}_{(\text{GSD and LST})}$ between the GSD of all points in the study area and the regional average LST of the study area, the distribution of $\text{PCC}_{(\text{GSD and LST})}$ in annual ground surface deformation rate (AGSDR) reflects the overall GSD characteristics of the study area. Further, to analyze the GSD characteristics of different IWPs and Aeroport, five Aeroport, six LCP, and six HCP regions were randomly selected by circles with a radius of 50 m for each region to have a similar number of deformation points in each region (Figure 2). To reduce the effect of individual deformation point on the GSD, we calculated the average GSD within each circle to represent each region. These GSD were then plotted separately as Aeroport, LCP, and HCP categories and analyzed in relation to LST.

### 3.6.5. Transect Selection

Two perpendicular transects across the study area were selected to analyze the spatial heterogeneity of GSD and the responses of IWPs to changes in environmental factors (elevation, Landsat-based NDVI, MNDWI, LST, and MODIS-based ALT) (Figure 2). The two transects are selected by the rules of study area coverage and representativeness. The transects reasonably covered the whole research area and well-represented the IWP areas, thermokarst lakes, and the Saskylakh Aeroport. The PCC method was selected for statical analyses. First, points were selected every 30 m on the line segment, with 300 points along the N–S transect and 227 points along the W–E transects. Then, attribute values were assigned to these selected points. Finally, the correlation of different attributes was calculated based on these points.

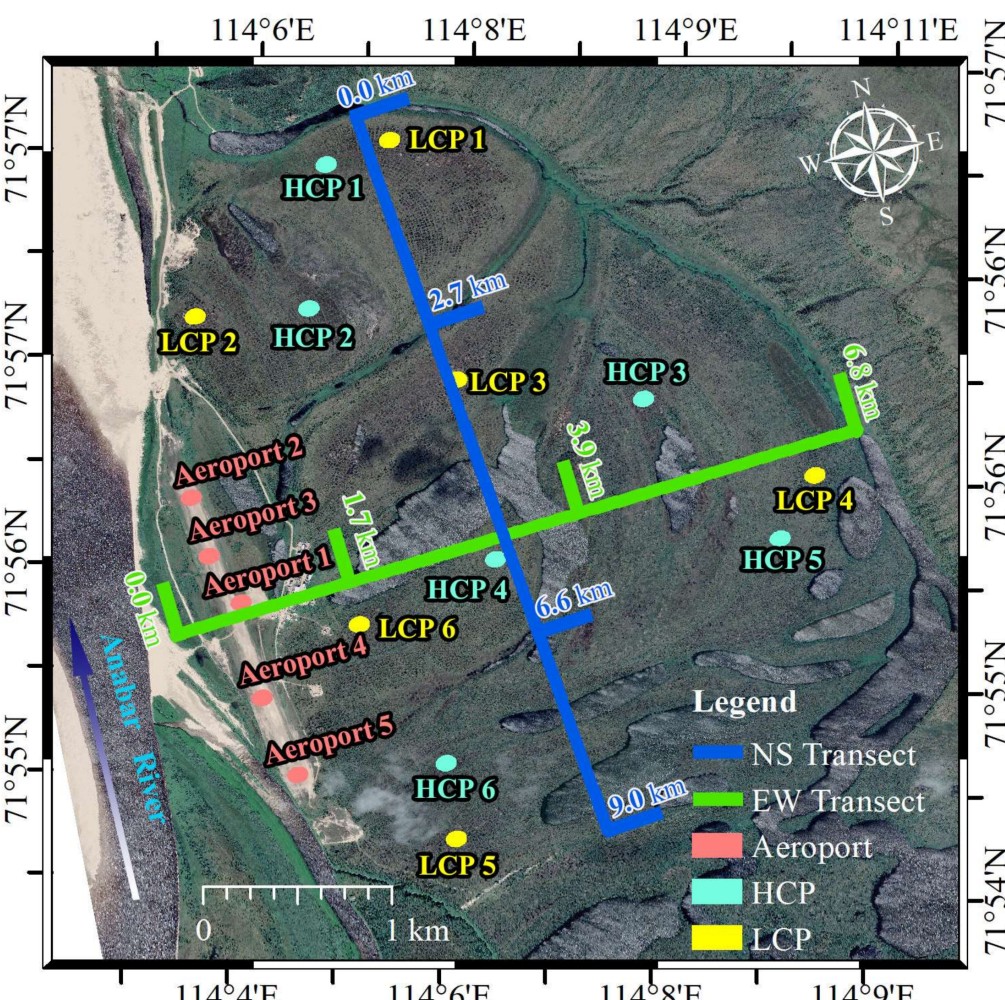

**Figure 2.** Locations of different types of ice-wedge polygons (IWPs) and N–S and W–E transects in Saskylakh, northwestern Yakutia, Siberian Arctic, Russia. Notes: (1) LCP: low-centered polygon; (2) HCP: high-centered polygon.

## 4. Results

### 4.1. SBAS-InSAR Accuracy Verification by Phase Variance

In this paper, the phase variance is used to verify the results of SBAS-InSAR. The larger the phase variance, the lower the accuracy, and; vice versa. The phase variance values of SBAS-InSAR range from 2.59 to 8.10 mm (Figure 3a), with those of most areas at 4–6 mm (Figure 3b). As shown in Figure 3a, the red parts are the areas with lower values of phase variance, and the blue parts are those with higher values of phase variance. As a good reflector, the runway terminal of the Saskylakh Aeroport changes a little over a long time and maintains a good relevance over a long period. As a result, the GSD of the Saskylakh Aeroport has a high accuracy in SBAS-InSAR results (Figure 3a). The SBAS-InSAR accuracy is lower in the areas with lush vegetation and near thermokarst ponds due to the low image coherence in these areas (Figure 3a).

### 4.2. GSD Rate in the Study Area and Different Areas of Ice-Wedge Polygons (IWPs)

The annual GSD rates (AGSDRs) of the study area in the period from 1 June 2018 to 3 May 2019 are provided in Figure 4. Spatially, the AGSDR values in Saskylakh ranged from −49.73 to 45.97 mm/a; and in 75% of the study area, AGSDRs fell between −10~10 mm/a. The AGSDR characteristics for different types of IWPs differed remarkably. In this paper, they are examined in detail.

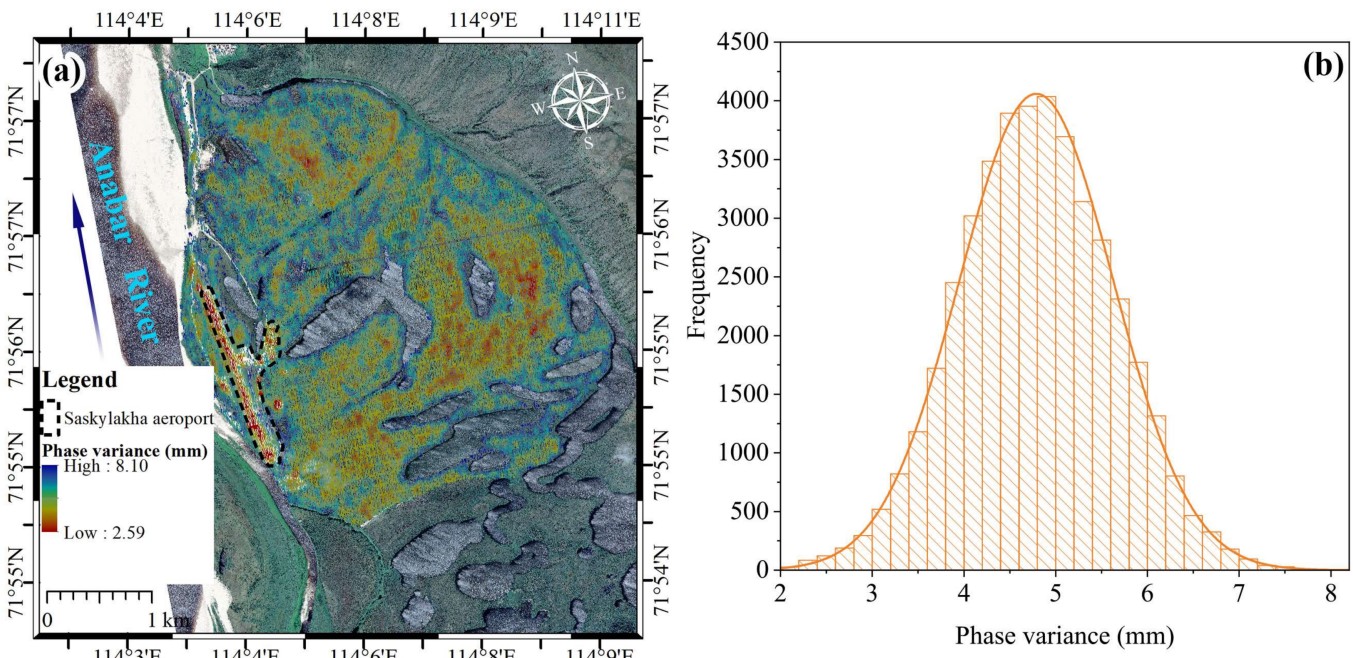

**Figure 3.** Phase variance distribution and histogram in Saskylakh, northwestern Yakutia, Siberian Arctic, Russia. Notes: (**a**) map of the spatial distribution of phase variance in SBAS-InSAR accuracy; and (**b**) histogram of phase variance in Saskylakh, in which the percentages in the figure are the proportion of all results accounted for by different phase variances.

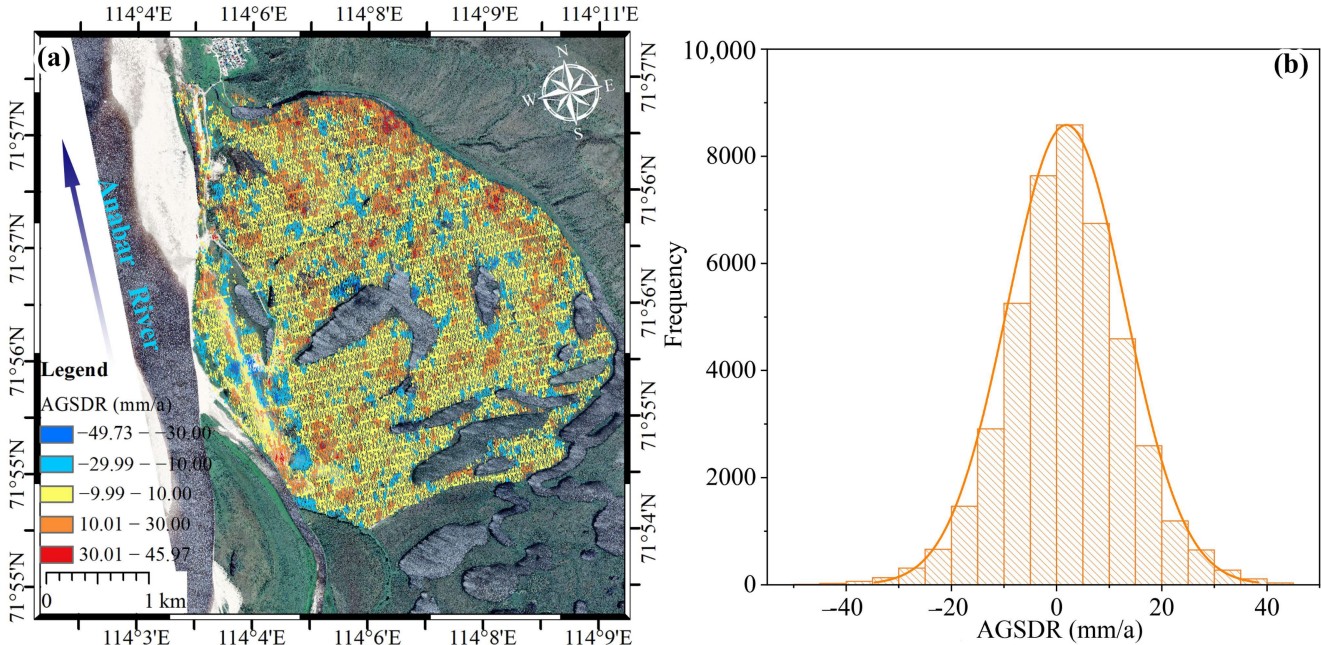

**Figure 4.** Annual ground surface deformation rate (AGSDR) and histogram of AGSDRs with frequency in Saskylakh, northwestern Yakutia, Siberian Arctic, Russia. Notes: (**a**) distribution of AGSDR in Saskylakh; and (**b**) histogram of AGSDR in Saskylakh, or the frequency distribution of percentage for different AGSDR values.

In general, the $PCC_{(GSD\ and\ LST)}$ is larger for points with larger AGSDR, and the $PCC_{(GSD\ and\ LST)}$ uncertainty is larger for AGSDR in −10~10 mm/a. The $PCC_{(GSD\ and\ LST)}$ is positive for points with negative AGSDR and negative for points with positive AGSDR (Figure 5a).

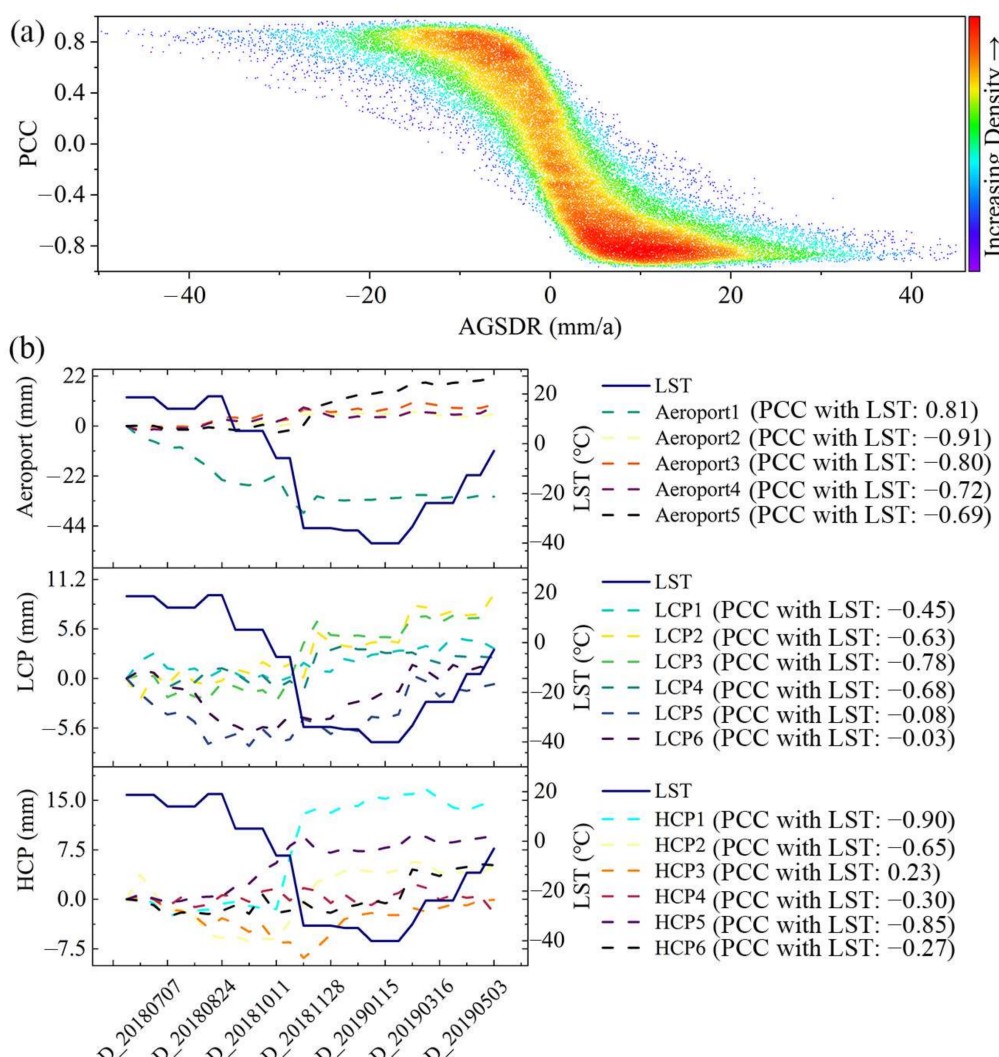

**Figure 5.** Ground surface deformation (GSD) of different IWP sites in representative areas in Saskylakh, northwestern Yakutia. Notes: (**a**) is the distribution of correlation between GSD and LST in annual ground surface deformation rate (AGSDR); (**b**) is the GST of LCP, HCP, and Aeroport in different regions. Notes for acronyms: LST: land surface temperature calculated by MODIS; LCP: low-centered polygon; HCP: high-centered polygon; PCC: Pearson correlation coefficient.

Further, the GSD characteristics of Aeroport, LCP, and HCP are also different. The GSD of the five Aeroport regions is larger than that of the LCP and HCP regions. Aeroport 1 is located in the tower of Aeroport, and the GSD shows a significant subsidence trend and is positively correlated with LST. The regions of Aeroport 2 to Aeroport 5 are located in the Aeroport runway; and the GSD shows different degrees of uplift and is negatively correlated with LST (Figure 5b). The GSDs of most LCPs and HCPs were negatively correlated with LST, and only the GSD of region HCP3 was weakly positively correlated with LST. The GSD of HCP was greater than the GSD of LCP (Figure 5b).

### 4.3. Ground Temperatures and Active Layer Thickness (ALT)

There is a certain pattern of changes in ground temperature. At greater depths, ground temperatures are subject to exponentially declining degrees of impact from climate change.

The modeled average ground temperatures and ALT in 1997–2020 were selected for analysis. Ground temperatures (GST, T1, and T10) were increasing gradually with fluctuations at Saskylakh from 1997 to 2019 (Figure 6a). The trend of ground warming varied with depths: the closer to the surface, the more rapidly the ground warming. For example, the increase rate of GST was 0.15 °C/a; and that of T10, 0.08 °C/a. Finally, the fluctuation amplitudes of ground temperatures also varied greater at the near-surface depths. This variability can be expressed by the range. The range of GST was 6.62 °C; and that of T10, 2.26 °C. In the last two decades, despite fluctuations, ALT has increased by 26 cm, with an average annual increasing rate of 0.94 cm/a (Figure 6b).

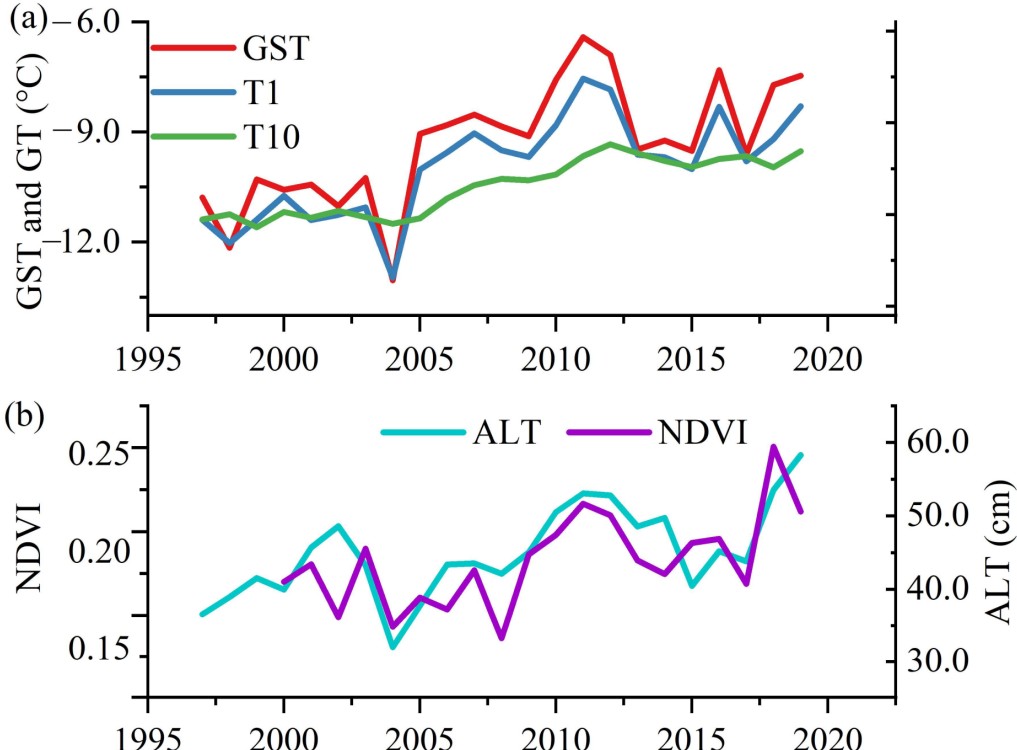

**Figure 6.** Dynamic of ground surface temperatures (GST) and ground temperatures (T1 and T10), active layer thickness (ALT) in 1997–2019 and NDVI in 2000–2020 at Saskylakh, Anabarsky District, the Sakha Republic, Russia. Notes: (**a**) shows the ground surface temperature at the depth of 5 cm (GST) and ground temperature at depths of 1 and 10 m (T1 and T10); (**b**) shows the normalized difference vegetation index (NDVI, −1~1) and active layer thickness (ALT).

### 4.4. Development of Thermokarst Lakes and Ponds

The thermokarst lakes/ponds below 1000 m$^2$ in the areal extent of the water surface were filtered according to image resolution. The areal extent of thermokarst ponds fluctuated from 2000 to 2020 (Figure 7a,b). The areal extent of thermokarst ponds was first expanding (2000–2010) and then shrinking (2010–2020). The areal change in thermokarst ponds may be mainly attributed to the short-term or seasonal water availability. It is worth noting that the areal extent of perennial thermokarst lakes and ponds was expanding (Figure 7a).

### 4.5. Permafrost Environment Reflected by Study Transects

Ice-wedge development and decay are related to ground and land surface temperatures and water/moisture availability. Ice-wedge development and decay are further complicated by vegetation and snow covers, the presence and disappearance of lakes and ponds, engineering construction and operation, land use and cover change, and soil properties under frozen and thawed states, among many others [61]. In this paper, we

revealed the distributive features of IWP areas near the Saskylakh Aeroport by studying the spatial heterogeneity of AGSDR and the responses of these features to changes in those environmental factors (elevation, NDVI, and MNDWI in study transects are the average values from 2000 to 2020, LST in transects is the average values from 2015 to 2020, and ALT) related to IWP, with two transects across the study area (Figure 2), which are:

(1) N–S direction transect (Figure 8). This is a 9 km-long N–S direction transect traversing across the study area and passing through the IWP fields (at the distance of 0.0~2.6, 2.7~4.4, 5.1~6.6, 7.0~7.5, 7.8~8.3, and 8.9~9.0 km), drainage ditches (at the distance of 2.6~2.7 km), and thermokarst lakes and ponds (at the distance 4.4~5.1, 6.6~7.0, 7.5~7.8, and 8.3~8.9 km).

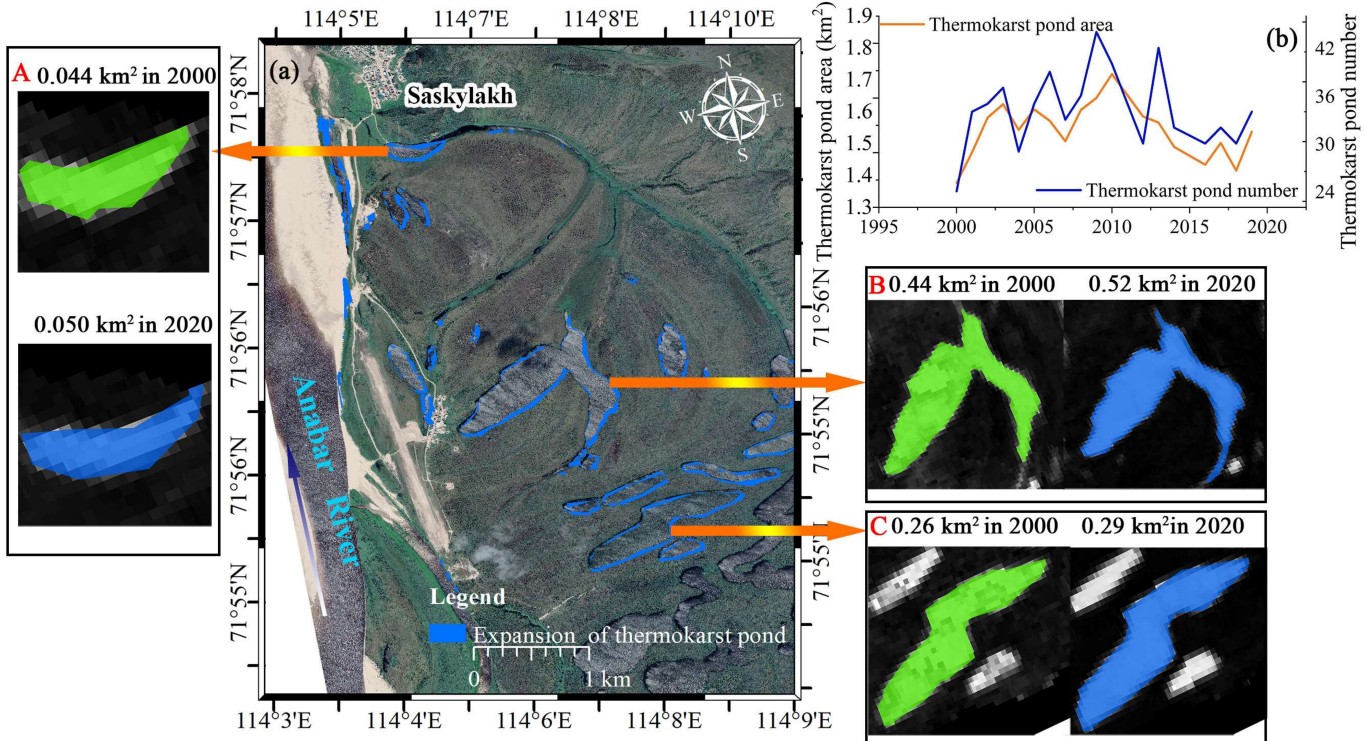

**Figure 7.** Expansion of perennial thermokarst lakes and ponds at Saskylakh, Anabarsky District, the Sakha Republic, Russia in 2000–2020 (The background Google Earth image was accessed in August 2013). Notes: (**a**) shows the expansion of perennial thermokarst lakes and ponds, (A), (B) and (C) are typical e expansion of perennial thermokarst lakes; (**b**) shows the fluctuation of thermokarst pond and number in 2000–2020.

(2) W–E direction transect (Figure 9). This is a 6.8 km-long W–E direction transect traversing across the study area and passing through the Saskylakh Aeroport (at the distance of 0.0~1.7 km), IWP (at the distance of 2.8~3.5, 3.9~4.7, and 5.2~6.8 km), and thermokarst lakes and ponds (at the distance of 1.7~2.8, 3.5~3.9, and 4.7~5.2 km).

It is found from the transects that: (1) there are only small elevation differences in the study area; (2) thermokarst lakes and ponds are mainly found on the low-lying terrains; (3) LST is lower in thermokarst lakes and ponds; (4) in the areas of thermokarst lakes and ponds, no deformation results were yielded due to the SAR data incoherence of water surfaces; (5) the active layer gradually thins northwards from 66.5 cm in the south to 56 cm in the north; (6) and the Saskylakh Aeroport is a representative of engineering disturbance, with high LST and low NDVI, and possibly large ALT.

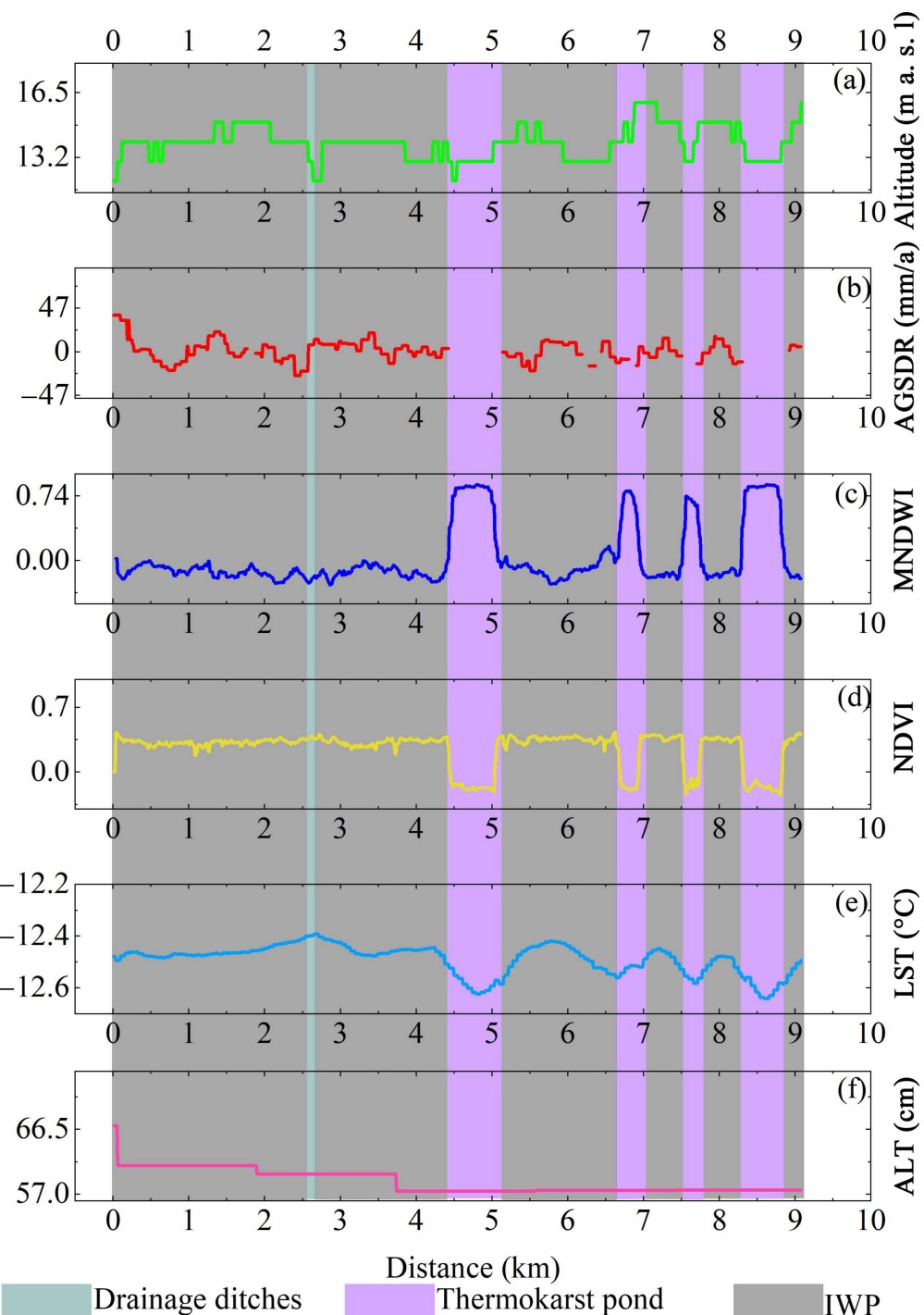

**Figure 8.** The N–S transect at Saskylakh, Anabarsky District, the Sakha Republic, Russia (The starting point of the N–S Transect is 0.0 km from the NS Transect in Figure 2); We canceled the bold font. Notes: (**a**) is the altitude (m a. s. l); (**b**) is the annual ground surface deformation rate (AGSDR, mm/a); (**c**) is the modified normalized difference water index (MNDWI, −1~1). For thermokarst pond: MNDWI > 0; (**d**) is the normalized difference vegetation index (NDVI, −1~1); (**e**) is the land surface temperature (LST, °C); and (**f**) is the active layer thickness (ALT, cm).

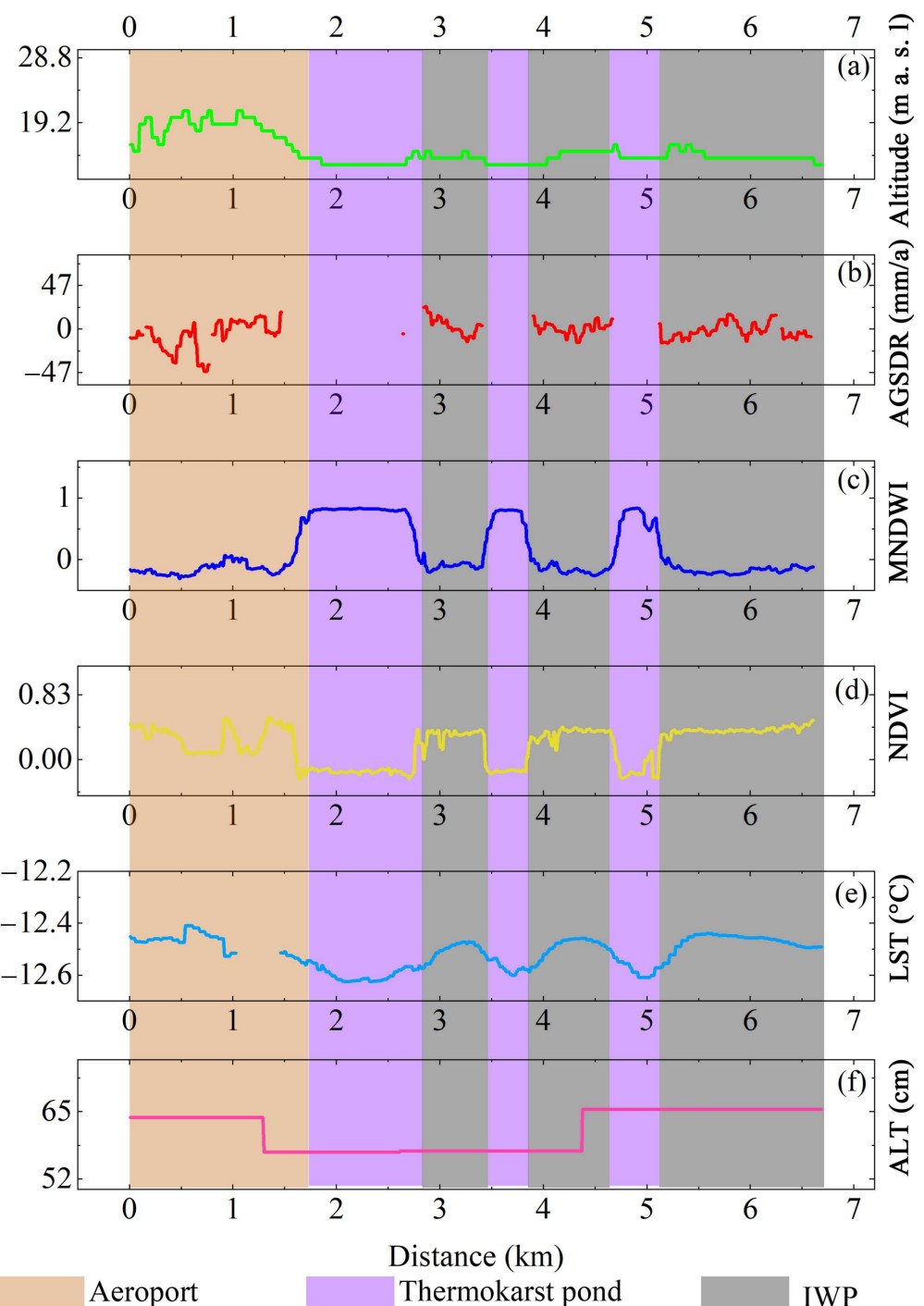

**Figure 9.** The W–E transect at Saskylakh, Anabarsky District, the Sakha Republic, Russia (the starting point of the W–E Transect is 0.0 km from the W–E Transect in Figure 2); Notes: (**a**) is the altitude (m a. s. l); (**b**) is the annual ground surface deformation rate (AGSDR, mm/a); (**c**) is the modified normalized difference water index (MNDWI, −1~1). For thermokarst pond: MNDWI > 0; (**d**) is the normalized difference vegetation index (NDVI, −1~1); (**e**) is the land surface temperature (LST, °C); and (**f**) is the active layer thickness (ALT, cm).

## 5. Discussions and Prospects

### 5.1. Changes in the Permafrost Environment

In recent years, ground temperatures and LST have been rising in the Arctic and the hydroclimate variability is on the rise. Higher ground temperatures and LST have led

to greater thaw depths, larger ALT, and lusher vegetation [62]. Permafrost in Saskylakh shows the characteristics of low LST and small ALT, but the vegetation coverage increases and the arctic greening is obvious. According to the prediction of Pearson et al. [62] the vegetation types in Siberia, Alaska, and other regions in the Arctic will change, and the range of shrubs and trees has expanded and will expand sharply. This paper also found that the characteristics of vegetation coverage increase in Saskylakh, including taller, denser, and lusher vegetation in the IWP areas, with rapidly and extensively emerging bushes and trees. In this study area at Saskylakh, NDVI has shown a fluctuating upward trend in the last 20 years (Figure 6); both NDVI and/or vegetation growth rates are greater than those on the Arctic Coastal Plain of northern Alaska at the same latitude [63]. Compared with the LST in areas at the same latitude, the variation range of LST of Saskylakh is smaller than that of Northern Québec, Nunavik, Canada, and of the North Slope of northern Alaska, USA [56]. Globally, ALT and its average annual rate of change in Saskylakh are relatively small. For example, ALT in Mongolia and on the Qinghai–Tibet Plateau is generally found between 100 and 300 cm, and the ALT in Nordic countries and the Alps even exceeds 300 cm [29].

In recent years, increasing human activities have affected the arctic eco-environment. In particular, the destruction or modifications of vegetation and the reduction in plant species diversity by infrastructure construction and operation also have further aggravated the permafrost degradation in the Arctic and Boreal zones [64,65]. In this study, the Saskylakh Aeroport under engineering impacts or other anthropogenic disturbances has severe GSD. A large range of settlement funnels are formed in the tower zone and a banded surface uplift is formed along the Aeroport runway. The elevational difference in the study area is less than 10 m. The major influences of elevation on the IWPs and the permafrost environment are reflected by the formation and expansion of thermokarst lakes and ponds on low-lying terrains. The latter is one of the important signals of IWP degeneration and permafrost degradation.

Studies have demonstrated that thermokarst activation and expansion are important indicators of IWP degeneration [15,66,67]. The areal extent of thermokarst lakes and ponds of Saskylakh shows a fluctuant change, increasing and then decreasing. A similar pattern was observed in a small area on the northeastern Qinghai–Tibet Plateau (QTP) but with a general shrinking trend of 41% for the period 1986–2015 [68]. The opposite on the central QTP was reported, a thermokarst expansion of 123% in 50 years [69]; while in northeastern Siberia, Kolyma lowland tundra, a thermokarst expansion of only 4.5% was recorded [70]. Because ice wedges are typically developed near the top of attached permafrost, polygonal terrain is particularly susceptible to thermokarsting. Thermokarst ponding affects the permafrost terrains when the thawing of ice-rich permafrost and the ensued meltwater drainage and enhanced soil erosion causes the loss of structural integrity of IWPs and the subsequent collapse of the ground [11,71–75].

### 5.2. GSD of Ice-Wedge Polygons (IWPs) and Permafrost Dynamics

The GSD of IWP areas is responsive to climate change [74]. The typical IWP regions and two transects show that in our study areas, with lowering ground temperature, the ground surface upheaved, and with rising ground temperature, subsided (Figures 5, 8 and 9). This is consistent with earlier findings in the Arctic. For example, after a decade of automated multi-sensor monitoring of LST in the Adventdalen Valley, central Svalbard, Norway, Matsuoka et al. [8] found that ground upheaving mainly occurred upon the ground freezing which generally started in late September to early November, and ground surface subsidence, upon the ground thawing, generally started in late May or early June.

The GSD in this paper is similar to the GSD of IWP areas monitored by Iijima et al. [75]. Both theirs and ours have achieved a wide range of monitoring, and cannot achieve accuracy of a single IWP research. Ground-based measurements allow for fine-grained studies. For example, after a multi-year (1996–2018) leveling study in the Mackenzie delta area, Canada,

Burn et al. [76] found the more intense settlement of ice-wedge troughs in comparison with that of IWPs. Similarly, we found that the deformation of ice-wedge troughs in the N–S transect (Figure 8) is larger than that in the surrounding areas. In addition, they found higher rates of GSD above ice wedges on hillslopes than those in other terrains because of the enhanced ground warming from downslope snow and wedge-ice meltwater runoff. It is worth noting that the GSD of Saskylakh Aeroport over arctic permafrost is more severe than that of natural ground surface under engineering disturbances. For example, the Saskylakh Airport has a large GSD; a settlement funnel has formed under and around the tower, and the GSD on the airport runway has a rising trend. The GSD of the Iqaluit Airport on the Baffin Island, Canadian Arctic also has similar characteristics, and the GSD of the airport runway is larger than that of the surrounding area [77].

*5.3. Prospects*

There are some inadequacies in this study, such as the limited penetration ability of the C-band Sentinel-1 satellite, the inconsistent resolution of optical satellites, the similar spatial resolution of data sources, and the lack of ground-based measurement support. Studies of changes in the permafrost ecological environment under a changing climate need long permafrost monitoring data, more diverse and reliable climate change indicators, and different spatial scales in research area. Due to the limitation of Sentinel-1 monitoring capability in the C-band, some indicators for permafrost degradation cannot be clearly reflected in GSD. Thus, in the future, longer time series of SAR images of higher resolution coupled with ground-based measurements are deemed necessary for validating and elaborating the results of InSAR-based studies on GSD of IWP areas.

The research scopes of different spatial scales should also be refined to different permafrost regions. More accurate monitoring data for frozen ground or talik of various types, and/or remote sensing inversion data, should be used to obtain more comprehensive climate change parameters. The degradation of IWP should be further validated and quantified by longer-term monitoring of borehole temperatures and ALT, as well as field surveys on hydroclimate, ecology, geocryology, and ecohydrology. Meanwhile, the characteristics and mechanisms of GSD of IWP areas with larger areal extents of standing water surfaces in the summer and lake ice extents in winter await further studies. In addition, the relationships between IWP and distributive features of surface and subsurface waters and vegetation and snow covers should be progressively refined.

## 6. Conclusions

In this paper, we present an one-year GSD study of IWP areas in Saskylakh, northwestern Yakutia, Russian Arctic by InSAR technique and GEE, and their influencing factors. The results indicate that the IWPs have been degrading under the joint action of several key elements. This study provides referential methods and findings to IWP processes and conditions in remote regions, where detailed ground-based observations are more difficult to obtain. We draw the following conclusions:

(1)  The AGSDR in Saskylakh ranged from −49.73 to 45.97 mm/a in 2018–2019, and in 75% of the study area, the AGSDR was between −10~10 mm/a;

(2)  All GSD points in the region indicated a closer relationship between GSD and LST for our observational points with larger AGSDR, and positive correlation between GSD and LST for those points with negative AGSDR, and vice versa;

(3)  Further, the GSDs of the five Aeroport areas indicated the most drastic deformation under engineering disturbances in the Saskylakh Aeroport area; and the GSDs of the six LCP and six HCP areas indicated that the GSDs of most LCPs and HCPs are negatively correlated with LST;

(4)  Arctic permafrost in Saskylakh showed a trend of degradation, as evidenced by rising ground temperature and NDVI (shown as denser and taller vegetation), a deepening active layer, and expanding thermokarst lakes and ponds;

(5)  Spatially, the higher the vegetation cover, the higher the LST and the larger the ALT.

**Author Contributions:** Conceptualization, W.W. and H.J.; Writing—Original Draft: W.W. and H.J.; Writing—Review and Editing: H.J., W.W., Z.Z., X.L., R.-D.Ș., Q.W., X.J. and M.Ș.; methodology, W.W. and Y.W.; formal analysis, A.L., S.Z. and V.T.; visualization, W.W., H.J. and S.Y.; supervision, M.N.Z. and V.V.S.; project administration, H.J.; funding acquisition, H.J., X.J., R.-D.Ș. and W.W. All authors have read and agreed to the published version of the manuscript.

**Funding:** This research was financially funded by the National Natural Science Foundation of China (NSFC Grant Nos. U20A2082 and 42101119), Joint Heilongjiang Province R&D and Northeast Forestry University Chengdong Leadership Program (Grant No. LJ2020-01), the Fundamental Research Funds for the Central Universities (Grant No. 2572021DT08 and 2572022AW53), and the Autonomous Province of Bozen/Bolzano—Department for Innovation, Research and University in the frame of the Seal of Excellence Programme (project TEMPLINK, Grant No. D55F20002520003). Details: (1) Key Program of NSFC Joint Foundation with Heilongjiang Province for Regional Development "Response mechanisms and carbon cycling processes of the Xing'an permafrost to global change" (Grant No. U20A2082); (2) NSFC Program "Shrubification and its hydrothermal impacts on the frozen ground along the China-Crude Oil Pipelines" (Grant No. 42101119); (3) NSFC Program "Mechanisms and trends of changes in southern/lower limits of latitudinal permafrost in Northeast China" (Grant No. 41871052). (4) Joint Chengdong Leadership and R&D Program of Heilongjiang Province and Northeast Forestry University "Degradation of permafrost in the Xing'anling Mountains and Outer Baikalia and adjacent regions and its engineering and environmental impacts" (Grant No. LJ2020-01). (5) Fundamental Research Funds for the Central Universities, "Stability and carbon release potential of permafrost carbon pools in the Da and Xiao Xing'anling Mountains under a warming climate" (Grant No. 2572021DT08). (6) Fundamental Research Funds for the Central Universities, "Water damage hazard monitoring in the zones of degrading permafrost alone the China-Russia Crude Oil Pipeline routes" (Grant No. 2572022AW53). (7) Raul-David Serban received funding from the Autonomous Province of Bozen/Bolzano—Department for Innovation, Research and University in the frame of the Seal of Excellence Programme (project TEMPLINK, Grant No. D55F20002520003).

**Data Availability Statement:** The data that support the findings of this study are available from the corresponding author upon reasonable request.

**Acknowledgments:** All authors are appreciative of the Sentinel-1 B images provided by the European Space Agency (ESA) and Alaska Satellite Facility (ASF), the digital elevation model (DEM) provided by Japan Aerospace Exploration Agency (JAXA), active layer thickness (ALT) and ground temperature data provided by the European Space Agency's (ESA) Climate Change Initiative (CCI) Permafrost project, meteorological station and land classification provided by Global Historical Climatology Network daily (GHCNd, URL: https://www.ncei.noaa.gov/products/land-based-station/global-historical-climatology-network-daily, accessed on 26 February 2023), and the Food and Agriculture Organization (FAO) of the United Nation (URL: https://www.fao.org/soils-portal/data-hub/soil-classification/en/, accessed on 26 February 2023). We are particularly grateful to Google Earth Engine (GEE) for the support of this paper.

**Conflicts of Interest:** The authors declare no known competing financial interest or personal relationships that could have appeared to influence the work reported in this paper.

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
