# Peer review of "Monitoring Ground Surface Deformation of Ice-Wedge Polygon Areas in Saskylakh, NW Yakutia, Using Interferometric Synthetic Aperture Radar (InSAR) and Google Earth Engine (GEE)"

_remotesensing, doi:10.3390/rs15051335_

Round 1
Reviewer 1 Report
Wang et al. used SBAS-InSAR and GEE investigated dynamics of ground surface deformation in ice-wedge polygon zones (2018-2019) and IWP-related environmental factors for influencing the IWPs in Saskylakh, northwestern Yakutia, Russia over the last 20 years. A deformation characteristics and changes in permafrost-related elements are summarized, the study provide valuable information about IWP processes and conditions for evaluating changes and paleo-reconstruction in the periglacial environment in the Arctic and Boreal zones. I think the article is suitable for publication in remote sensing. However, the article needs a minor revision.
My major comments are listed below.
1. In the description of the study location in Study area and sites in Saskylakh, the eastern part of the study area is not mentioned, please improve it.
2. Line 141: Please change Google Earth 9.168.0.0 to image accessed time.
3. Lines 142 and 145: Please change insets to inseted.
4. Line 163: In Table 2, Landsat 7 ETM+ and Landsat 8 OLI/TIRS are used for LST or NDVI and MNDWI, Landsat 7 ETM+ and Landsat 8 OLI/TIRS are need to separate descriptions.
5. In Data and methods, 3.2- 3.4 have same title, I think it is a typographical error, please fix it.
6. Line 259: Time resolution needs to be correctly described.
7. Lines 339-348: The description begins with a summary statement and then develops in detail.
8. Please mark the location of Saskylakh in Figure 7.
9. Line 373: There is no discussion in section 5, please confirm it.
10. Line 438: There is a typographical error.
11. In the Conclusion and prospects, there is no content of remote sensing monitoring of permafrost-related elements, and it is suggested to add this content.
Author Response
Dear Editors and Reviewers,
We thank the Remote Sensing editors and reviewers for their kind and valuable comments and advice on this article. After carefully checking the expert opinions, we revised the paper point-to-point according to the expert opinions and responded to the raised concerns and revision opinions. Under each question, the revised part of the manuscript is marked in red. The following are the expert opinions we revised and answered point-to-point.
Reviewer 1:
Wang et al. used SBAS-InSAR and GEE investigated dynamics of ground surface deformation in ice-wedge polygon zones (2018-2019) and IWP-related environmental factors for influencing the IWPs in Saskylakh, northwestern Yakutia, Russia over the last 20 years. A deformation characteristics and changes in permafrost-related elements are summarized, the study provide valuable information about IWP processes and conditions for evaluating changes and paleo-reconstruction in the periglacial environment in the Arctic and Boreal zones. I think the article is suitable for publication in remote sensing. However, the article needs a minor revision.
My major comments are listed below.
Q1: In the description of the study location in Study area and sites in Saskylakh, the eastern part of the study area is not mentioned, please improve it.
Authors’ Responses:
We added the description of eastern part of the study area in subsection 2. Study area and sites in Saskylakh.
Revised manuscript:
Lines 106-107:
The eastern side is a terrace with sparse vegetation, which is nearly 20 m higher than the study area.
Q2: Line 141: Please change Google Earth 9.168.0.0 to image accessed time.
Authors’ Responses:
We changed “ Google Earth 9.168.0.0” to image accessed time in Figure 1.
Revised manuscript:
Lines 150-151:
(c) Study area image (The background image is a Google Earth image, accessed in August 2013); the yellow line is the study boundary, and;
Q3: Lines 142 and 145: Please change insets to inseted.
Authors’ Responses:
Disagreed. We believe that the word “insets” here in Figure 1 has been properly used. Inset can be used either in verb or noun form. Using inset here means one of the maps inset or inserted in the main map or a combo of maps put together.
Q4: Line 163: In Table 2, Landsat 7 ETM+ and Landsat 8 OLI/TIRS are used for LST or NDVI and MNDWI, Landsat 7 ETM+ and Landsat 8 OLI/TIRS are need to separate descriptions.
Authors’ Responses:
We describe Landsat 7 and Landsat 8 separately in Table 2 Optical and DEM data parameters used in this paper.
Revised manuscript:
Line: 180:
Table 2. Optical and DEM data parameters used in this paper
Sensor |
Spatial resolution |
Parameter |
Bands |
Unit |
Range |
Value type |
Landsat 8 OLI/TIRS |
30 m |
LST |
B4, B5, B10 |
℃ |
/ |
Year-Average |
NDVI |
B4, B5 |
/ |
-1 ~ 1 |
Year-Average |
||
MNDWI |
B3, B6 |
/ |
-1 ~ 1 |
Year-Average |
||
Landsat 7 ETM+ |
NDVI |
B3, B4 |
/ |
-1 ~ 1 |
Year-Average |
|
MNDWI |
B2, B5 |
/ |
-1 ~ 1 |
Year-Average |
||
MODIS (MOD11A1) |
1,000 m |
LST |
/ |
℃ |
/ |
Month- Average |
ALOS |
30 m |
DEM |
/ |
m |
/ |
/ |
Q5: In Data and methods, 3.2- 3.4 have same title, I think it is a typographical error, please fix it.
Authors’ Responses:
We modified subsection 3.2-3.4 and renamed them.
Revised manuscript:
Line 169:
3.2. Optical data
Line 181:
3.3. Permafrost data
Line 199:
3.4. SBAS-InSAR
Q6: Line 259: Time resolution needs to be correctly described.
Authors’ Responses:
We corrected the time resolution in line 259 (modified in Lines 283-284).
Revised manuscript:
Lines 284-286:
This paper uses LST data with two different spatial and temporal resolutions: the Landsat 8 OLI/TIRS data with 30-m spatial resolution and a 16-day revisit period and the MODIS data with 1000-m spatial resolution and an one-day revisit period.
Q7: Lines 339-348: The description begins with a summary statement and then develops in detail.
Authors’ Responses:
We added a summary statement in Subsection 4.3 Ground temperatures and active layer thickness (ALT).
Revised manuscript:
Lines 392-395
There is a certain pattern of ground temperature change. At greater depths, ground temperatures are subject to exponentially declining degrees of impacts from climate change. Areal average ground temperatures and ALT in 1997-2020 were selected for analysis.
Q8: Please mark the location of Saskylakh in Figure 7.
Authors’ Responses:
We mark the location of Saskylakh in Figure 7.
Revised manuscript:
Lines 425-427:
Figure 7. Expansion of thermokarst lakes and ponds at Saskylakh, Anabarsky District, the Sakha Republic, Russia in 2000-2020 (The background Google Earth image was accessed in August 2013).
Q9: Line 373: There is no discussion in section 5, please confirm it.
Authors’ Responses:
We have elaborated prospects accordingly in 6 Conclusion and prospects.
Revised manuscript:
Lines 568-587:
There are some shortcomings in this study, such as the limited penetration ability of the C-band Sentinel-1 satellite, the inconsistent resolution of optical satellites, the unified resolution of data sources, and the lack of ground-based measurement support. Studies of changes in the permafrost ecological environment under a changing climate needs long permafrost monitoring data, more climate change indicators, and different spatial scales in research area. In future studies, due to the limitation of Sentinel-1 monitoring capability in the C-band, some indicators for permafrost degradation cannot be clearly reflected in GSD. Thus, longer time series of SAR images of higher resolution coupled with ground-based measurements are deemed necessary for validating and elaborating the results of InSAR surface deformation of IWP areas.
The research scope of different spatial scales should be refined to different permafrost regions. More accurate frozen ground or talik monitoring data or remote sensing inversion data should be used to obtain more comprehensive climate change parameters. The degradation of IWP should be further validated and quantified by longer-term monitoring of borehole ground temperatures and ALT, as well as field surveys hydroclimate, ecology, and ecohydrology. Meanwhile, the characteristics and mechanisms of GSD of IWP areas with a larger areal extent of standing water surfaces in summer and lake ice extents in winter await further studies. In addition, the relationships between IWP and distributive features of surface and subsurface waters and vegetation and snow covers should be progressively refined.
Q10: Line 438: There is a typographical error.
Authors’ Responses:
We corrected the typographical error.
Before revise:
The GSD? of dry HCP and LCP are negatively correlated with LST, while that of the wet HCP and LCP are positively correlated with LST.
After revise:
Lines 513-514:
The GSD of dry HCP and LCP are negatively correlated with LST, while that of the wet HCP and LCP are positively correlated with LST.
Q11: In the Conclusion and prospects, there is no content of remote sensing monitoring of permafrost-related elements, and it is suggested to add this content.
Authors’ Responses:
We added content of remote sensing monitoring of permafrost-related elements in 6 Conclusion and prospects.
Lines 570-573:
Studies of changes in the permafrost ecological environment under a changing climate needs long permafrost monitoring data, more climate change indicators, and different spatial scales in research area.
Lines 579-580:
More accurate frozen ground or talik monitoring data or remote sensing inversion data should be used to obtain more comprehensive climate change parameters.

Reviewer 2 Report
see attachment

Author Response
Dear Editors and Reviewers,
We thank the Remote Sensing editors and reviewers for their kind and valuable comments on this article. After carefully checking the expert opinions, we have revised the paper point-to-point according to the expert opinions and have answered all the raised opinions. The revised manuscript is marked under each question in red. The following are the expert opinions we revised and answered point-to-point.

Reviewer 3 Report
This study investigated the surface deformation of ice-wedge polygons in the Saskylakh region using short-baseline subset InSAR technology and analyzed the environmental factors associated with the ice wedge polygons. This paper achieved better results of surface deformation monitoring and environmental factors analysis of ice wedge polygons. I think this paper is valuable for the study of Arctic permafrost.
In my view, this paper has value to publish in Remote Sensing, but it still needs a minor revision before publication.
1. The section titles of sections 3.2, 3.3 and 3.4 are the same, please confirm.
2. Please give the full name of MODIS in line 163.
3. There is a misspelled sentence in line 269.
4. As shown by the titles of sections 5 and 6, both two sections contained prospects, please confirm.
5. There is a less of discussion of the elevation data used in section 5.
6. There is a misspelled sentence in line 440.
7. The paper's writing still needs a lot of improvement.
Author Response
Dear Editors and Reviewers,
We thank the Remote Sensing editors and reviewers for their kind and valuable comments on this article. After carefully checking the expert opinions, we revised the paper point-to-point according to the expert opinions and answered the raised opinions. The revised manuscript is marked under each question in red. The following are the expert opinions we revised and answered point-to-point.
Reviewer 3:
This study investigated the surface deformation of ice-wedge polygons in the Saskylakh region using short-baseline subset InSAR technology and analyzed the environmental factors associated with the ice wedge polygons. This paper achieved better results of surface deformation monitoring and environmental factors analysis of ice wedge polygons. I think this paper is valuable for the study of Arctic permafrost.
In my view, this paper has value to publish in Remote Sensing, but it still needs a minor revision before publication.
Q1: The section titles of sections 3.2, 3.3 and 3.4 are the same, please confirm.
Authors’ Responses:
We modified subsection 3.2-3.4 and renamed them.
Revised manuscript:
Line 169:
3.2. Optical data
Line 181:
3.3. Permafrost data
Line 199:
3.4. SBAS-InSAR
Q2: Please give the full name of MODIS in line 163.
Authors’ Responses:
We added the full name of MODIS.
Revised manuscript:
Lines 173:
……, and Moderate Resolution Imaging Spectroradiometer (MODIS).
Q3: There is a misspelled sentence in line 269.
Authors’ Responses:
We corrected misspelled sentence in line 269.
Revised manuscript:
Lines 333-334:
As shown in Figure 3a, the red parts are the areas with lower values of phase variance, and; the blue parts, those with higher values of phase variance.
Q4: As shown by the titles of sections 5 and 6, both two sections contained prospects, please confirm.
Authors’ Responses:
We have unified prospects on 6 Conclusion and prospects.
Revised manuscript:
Lines 568-587:
There are some shortcomings in this study, such as the limited penetration ability of the C-band Sentinel-1 satellite, the inconsistent resolution of optical satellites, the unified resolution of data sources, and the lack of ground-based measurement support. Studies of changes in the permafrost ecological environment under a changing climate needs long permafrost monitoring data, more climate change indicators, and different spatial scales in research area. In future studies, due to the limitation of Sentinel-1 monitoring capability in the C-band, some indicators for permafrost degradation cannot be clearly reflected in GSD. Thus, longer time series of SAR images of higher resolution coupled with ground-based measurements are deemed necessary for validating and elaborating the results of InSAR surface deformation of IWP areas.
The research scope of different spatial scales should be refined to different permafrost regions. More accurate frozen ground or talik monitoring data or remote sensing inversion data should be used to obtain more comprehensive climate change parameters. The degradation of IWP should be further validated and quantified by longer-term monitoring of borehole ground temperatures and ALT, as well as field surveys hydroclimate, ecology, and ecohydrology. Meanwhile, the characteristics and mechanisms of GSD of IWP areas with a larger areal extent of standing water surfaces in summer and lake ice extents in winter await further studies. In addition, the relationships between IWP and distributive features of surface and subsurface waters and vegetation and snow covers should be progressively refined.
Q5: There is a less of discussion of the elevation data used in section 5.
Authors’ Responses:
We added discussion about the elevation in Subsection 5.1 Changes in the permafrost environment.
Revised manuscript:
Lines 488-491:
The elevational difference in the study area is less than 10 m, which has little influence on IWPs and permafrost. Their major influences on the IWPs and the permafrost environment are reflected by the formation and expansion of thermokarst lakes and ponds on the low-lying terrains.
Q6: There is a misspelled sentence in line 440.
Authors’ Responses:
We corrected the sentence in line 440.
Revised manuscript:
Before revise:
The GSD? of dry HCP and LCP are negatively correlated with LST, while that of the wet HCP and LCP are positively correlated with LST.
After revise:
Lines 513-514:
The GSD of dry HCP and LCP are negatively correlated with LST, while that of the wet HCP and LCP are positively correlated with LST.
Q7: The paper's writing still needs a lot of improvement.
Authors’ Responses:
We have revised the grammar of the whole article and focused on the use of singular/simple, so as to reduce the grammatical problems of the article. There are too many modifications, which are displayed in red font in the article.

Reviewer 4 Report
The title of the paper requires correction. Polygon is singular, and you studied polygons, or polygonal areas. Another issue is a role of GGE in you studies. From the text it is clear that GEE was used to work with optical data. The information about software you used to work with SAR data is missing. The main methodology flaw, from my point, is resolution issue. The tipical polygon size in the area are close to 5x5 m. The resolution of SAR data is 5x20 m, optical data for lake surface calculation - 30 m, LST, ALT - 1000 m. From this perspective, the accuracy of profiles you comparing looks not high.
You are operating the annual dynamics GSD numbers, but have only 1 year of observation, which is to short period. As I know, the Sentinel images are available from 2015, why not to use longer series?
Why Saskylakh? Choosing the study area should be explained in the text. You have no ground validation measurments here, why not to choose better equiped area?
line 89 - excess dot after (GEE). The sentence is very long, please rephrase.
Figure 1 - the meaning of a-d is missing in notes, and numbers (1-5) are missing on the image. Scale bar at insert d is highly appreciated.
line 166 - wrong title
line 181 - wrong title
line 237 - 3.6.1
line 244 - 3.6.2
line 254 - 3.6.3
Figure 4 - would be great to show contours of MODIS pixels, to understand how thew cover the observation points. If you use only values for one pixel per study area, please mark it on the map. Another point, as you defenetly cant measure the GSD for individual polygon, better to show areas with different polygon types, not single points.
Figure 6 - as ground temperatures are moddeled, not measured, I propose to show less depth (T1 and 10), at the moment the drawing is overloaded. Not all data are of the same resolution - the number of lakes are for study area, while other data are for one MODIS pixel (if I understand it correctly).
Figure 7 - would be better to use image from 2010 (year of maximum extend) as background, as you showing the difference.
Figure 8,9 - LST data looks too detailed, compared with 1000 m resolution of MODIS dataset, please clarify.
line 440 - excess ?
Figure 10 - as one observe on the figure, the correlation between GSD and LST is in yellow zone which means low. This is controversal to text.
Conclusions requires some qlarification. According to you results, the 75% or the study area experienced no significant GSD (-1...1 mm per year). The high numbers for aerport area are clearly related with antropogenic influence. Some degradation of permafrost is obvious, but seems like not reflected clearly in GSD, especially when you have only 1 year measurments.
This paper is a mix bag - the long time series of coarse resolution data about surfase temperatures was compared with one year of fine resolution data about surface deformations. The capabilities of the InSAR method to measure the ground deformation are well known, this paper have no ground validation, so, only average overal merit.
Author Response
Dear Editors and Reviewers,
We thank you for your kind and valuable comments and advice on this article. After carefully checking these expert opinions, we have revised the paper point-to-point accordingly and answered the raised questions. Under each question, the revised parts of the manuscript are marked out in red. The following are the expert opinions we have revised and answered point-to-point.
Reviewer 4:
Q1: The title of the paper requires correction. Polygon is singular, and you studied polygons, or polygonal areas. Another issue is a role of GGE in you studies. From the text it is clear that GEE was used to work with optical data. The information about software you used to work with SAR data is missing. The main methodology flaw, from my point, is resolution issue. The tipical polygon size in the area are close to 5x5 m. The resolution of SAR data is 5x20 m, optical data for lake surface calculation - 30 m, LST, ALT - 1000 m. From this perspective, the accuracy of profiles you comparing looks not high.
Authors’ Responses:
Thanks for your comments.
We modified the paper title to Monitoring ground surface deformation of ice wedge polygon areas in Saskylakh, Northwest Yakutia, using Interferometric Synthetic Aperture Radar (InSAR) and Google Earth Engine (GEE).
In subsection 3.4. SBAS-InSAR, we added the contents on the SBAS-InSAR processing platform.
The resolution of the data used in this paper is limited. Thus, we have not discussed a single IWP. We have studied different types of IWP to avoid the problem of insufficient resolution. The resolution of ALT distribution is 1000 m. Therefore, we have chosen the average value of the region to maximize the use of all data during the discussion.
Revised manuscript:
Lines 2-5:
Monitoring ground surface deformation of ice wedge polygon areas in Saskylakh, NW Yakutia, using Interferometric Synthetic Aperture Radar (InSAR) and Google Earth Engine (GEE)
Lines 211-212:
In this study, the SBAS-InSAR analysis was completed in SARscape 5.2.1.
Q2: You are operating the annual dynamics GSD numbers, but have only 1 year of observation, which is to short period. As I know, the Sentinel images are available from 2015, why not to use longer series?
Authors’ Responses:
We also tried in working out a longer time series, but because of the poor coherence of Sentinel-1 B SAR images in this region, the interference results were poor. This may have led to the failure of the experiment. Thus, we chose a period of one year to do the interference, successfully obtaining the results of this paper.
Q3: Why Saskylakh? Choosing the study area should be explained in the text. You have no ground validation measurments here, why not to choose better equiped area?
Authors’ Responses:
In the Introduction, we have explained the reasons for choosing Saskylakh as the research area. We chose this area as the research area because it is key for the arctic permafrost environment but otherwise rarely studied, either by Russian scientists or international scholars, and we hope to lay some good foundation so as to carry out more ground research work in the future.
Revised manuscript:
Lines 86-89:
Saskylakh is one of the key towns in the Siberian Arctic where the man-made surfaces are in good contrast with different types of IWP and thermokarst lakes and ponds. Therefore, the geo-cryological study in Saskylakh enables a comparative study of IWP and technogenic surfaces.
Q4: line 89 - excess dot after (GEE). The sentence is very long, please rephrase.
Authors’ Responses:
We rephrased the paragraph, and shorten it for easier understanding.
Revised manuscript:
Lines 97-101:
……;specific factors are digital elevation model (DEM), annual GSD rate (AGSDR) based SBAS-InSAR, land surface temperature (LST), modified normalized difference water index (MNDWI), normalized difference vegetation index (NDVI) based Landsat 8 OLI/TIRS, Landsat 7 ETM+, and active layer thickness (ALT) based permafrost CCI 2.3.
Q5: Figure 1 - the meaning of a-d is missing in notes, and numbers (1-5) are missing on the image. Scale bar at insert d is highly appreciated.
Authors’ Responses:
We rephrased the notes of Figure 1 to improve the note and insert a scale bar in d.
Revised manuscript:
Lines143-156:
Figure 1. Location of the study area in Saskylakh, northwestern Yakutia, Russia.
Notes:
(a) Global permafrost distribution and study area location, Saskylakh, Anabarsky District, the Sakha Republic, Russia (URL: https://climate.esa.int/en/projects/permafrost/) cited from [28];
(b) Location of the study area in the Anabar basin (URL: https://www.hydrosheds.org/);
(c) Study area image (The background image is a Google Earth image, accessed on August 2013); the yellow line is the study boundary, and;
(d) Representative examples of different IWPs. Insets (1) to (4) are the representative sample from Google earth images of HCP, Sparse vegetation LCP (SV-LCP), HCP Lush vege-tation LCP (LV-LCP), Sparse vegetation HCP (SV-HCP), and Lush vegetation HCP (LV-HCP) in the study area, and; Insets (d5) and (d6) are the representative samples of Google Earth images for the Saskylakh Aeroport and tundra, respectively.
Q6: line 166 - wrong title
line 181 - wrong title
Authors’ Responses:
We renamed the titles of 3.3-3.4 to 3.3. Permafrost data and 3.4. SBAS-InSAR.
Revised manuscript:
Line 181:
3.3. Permafrost data
Line 199:
3.4. SBAS-InSAR
Q7: line 269 - 3.6.1
line 276 - 3.6.2
line 286 - 3.6.3
Authors’ Responses:
We corrected line 262 to 3.6.1, line 269 to 3.6.2, and line 282 to 3.6.3.
Revised manuscript:
Line 259:
3.6.1. Normalized difference vegetation index (NDVI)
Line 266:
3.6.2. Modified normalized difference water index (MNDWI)
Line 279:
3.6.3. Land surface temperature (LST)
Q8: Figure 4 - would be great to show contours of MODIS pixels, to understand how thew cover the observation points. If you use only values for one pixel per study area, please mark it on the map. Another point, as you defenetly cant measure the GSD for individual polygon, better to show areas with different polygon types, not single points.
Authors’ Responses:
Due to the low resolution of MODIS, contour map cannot be formed, so contour map is not used in this paper.
We added the conditions of the sites selected, and we selected IWP points according to GSD, NDVI, IWP types, and MNDWI. Each IWP point has a large range of GSD, with large MDNWI differences between dry and wet IWP and significant NDVI differences between sparse and lush vegetation. Therefore, the sites we have selected are representative. Thus, we followed the typical point.
Revised manuscript:
Lines 295-311:
3.6.4 Selection of ice wedge polygon (IWP) points
The GSD characteristics of each type of IWP were studied at two different locations selected based on NDVI and Google Earth images, respectively (Figure 2). These points can represent different types of IWPs. We selected IWP points according to GSD, NDVI, IWP types, and MNDWI. Each IWP point has a large range of GSD, with large MDNWI differences between dry and wet IWP and large NDVI difference between sparse and lush vegetation. These points are representative for each IWP. Each IWP with different charac-teristics is guaranteed to have one representative point, in addition to the Saskylakh Aeroport and tundra point for comparisons with the IWP areas. Therefore, a total of ten points were selected. Those symbols of SV-LCP1, LV-LCP1, SV-HCP1, and LV-HCP1 rep-resent the areas with less surface water (dry LCP and HCP, MNDWI≤0; Figure 2), and those of SV-LCP2, LV-LCP2, SV-HCP2, and LV-HCP2, with more surface waters (wet LCP and HCP, MNDWI>0; Figure 2). The Saskylakh Aeroport (Figure 2) and nearby tundra (Figure 2) are adopted as the non-IWP sites for comparison. Two typical areas are selected for each IWP type to study the GSD characteristics of different IWP types. GSD in perma-frost areas is strongly related to variations in LST and ground ice contents [58-60]. In this paper, the GSD characteristics were compared with MODIS-based LST data.
Q9: Figure 6 - as ground temperatures are moddeled, not measured, I propose to show less depth (T1 and 10), at the moment the drawing is overloaded. Not all data are of the same resolution - the number of lakes are for study area, while other data are for one MODIS pixel (if I understand it correctly).
Authors’ Responses:
Agreed and done.
This data set has been verified by nearly 1000 borehole data in the world, and compared with other data sets. It is found that the accuracy of this data set is reliable. We adopted the comments of reviewers and used LST, T1 and T10 in Figure 6.
These data in Figure 6 take the average value of the region.
Revised manuscript:
Lines 193-198:
The permafrost data provided by permafrost CCI 2.3 were verified by 920 boreholes from the Global Terrestrial Network for Permafrost (GTN-P) and the Thermal State of Perma-frost (TSP) project network and from China. The permafrost data provided by permafrost CCI 2.3 are highly consistent, especially ALT [29], with slight deviations in ground tem-perature.
Lines 394-395:
Areal average ground temperatures and ALT in 1997-2020 were selected for analysis.
Lines 403-411:
Figure 6. Dynamic of ground surface temperatures (GST) and ground temperatures (T1 and T10), active layer thickness (ALT) in 1997-2019 and NDVI in 2000-2020 at Saskylakh, Anabarsky District, the Sakha Republic, Russia.
Notes:
1) GST stands for the ground surface temperature at the depth of 5 cm;
2) T1 and T10 stand for the ground temperature at depths of 1 m and 10 m;
3) NDVI, normalized difference vegetation index (-1 ~ 1), and;
4) ALT, active layer thickness.
Q10: Figure 7 - would be better to use image from 2010 (year of maximum extend) as background, as you showing the difference.
Authors’ Responses:
Thank you for your kind advice.
We want to use the high-resolution image of 2010, but only the high-resolution image of 2013 is available.
Q11: Figure 8,9 - LST data looks too detailed, compared with 1000 m resolution of MODIS dataset, please clarify.
Authors’ Responses:
Compared with other data, the resolution of ALT/MODIS is low, because our research is not based on a single IWP, but on a large range of IWP research, so the low resolution of ALT/MODIS has little impact on this study.
Q12: line 440 - excess ?
Authors’ Responses:
The word is not found in line 440, which is confused.
Q13: Figure 10 - as one observe on the figure, the correlation between GSD and LST is in yellow zone which means low. This is controversal to text.
Authors’ Responses:
After inspection, we found that there was a problem in the drawing, so we redraw Figure 10.
Revised manuscript:
Lines 525-535:
Figure 10. Correlation matrix of N-S and W-E transect ground surface deformation and IWP environment related factors.
Notes: (a) Correlation matrix of the N-S transect, and; (b) Correlation matrix of the W-E transect.
Acronyms:
- AGSDR stands for annual ground surface deformation rate (mm/a);
- MNDWI, modified normalized difference water index (-1~1), thermokarst pond: MNDWI>0;
- NDVI, normalized difference vegetation index (-1 ~ 1);
- LST, land surface temperature (℃), and;
- ALT, active layer thickness (cm).
Q14: Conclusions requires some qlarification. According to you results, the 75% or the study area experienced no significant GSD (-1...1 mm per year). The high numbers for aerport area are clearly related with antropogenic influence. Some degradation of permafrost is obvious, but seems like not reflected clearly in GSD, especially when you have only 1 year measurments.
Authors’ Responses:
InSAR technology is really difficult to achieve all ground surface deformation monitoring, and our monitoring time is relatively short. Thus, we rewrote the conclusion and discussion parts in order to improve these problems in future research.
Revised manuscript:
Lines 568-587:
There are some shortcomings in this study, such as the limited penetration ability of the C-band Sentinel-1 satellite, the inconsistent resolution of optical satellites, the unified resolution of data sources, and the lack of ground-based measurement support. Studies of changes in the permafrost ecological environment under a changing climate needs long permafrost monitoring data, more climate change indicators, and different spatial scales in research area. In future studies, due to the limitation of Sentinel-1 monitoring capability in the C-band, some indicators for permafrost degradation cannot be clearly reflected in GSD. Thus, longer time series of SAR images of higher resolution coupled with ground-based measurements are deemed necessary for validating and elaborating the results of InSAR surface deformation of IWP areas.
The research scope of different spatial scales should be refined to different permafrost regions. More accurate frozen ground or talik monitoring data or remote sensing inversion data should be used to obtain more comprehensive climate change parameters. The degradation of IWP should be further validated and quantified by longer-term monitoring of borehole ground temperatures and ALT, as well as field surveys hydroclimate, ecology, and ecohydrology. Meanwhile, the characteristics and mechanisms of GSD of IWP areas with a larger areal extent of standing water surfaces in summer and lake ice extents in winter await further studies. In addition, the relationships between IWP and distributive features of surface and subsurface waters and vegetation and snow covers should be progressively refined.
Q15: This paper is a mix bag - the long time series of coarse resolution data about surfase temperatures was compared with one year of fine resolution data about surface deformations. The capabilities of the InSAR method to measure the ground deformation are well known, this paper have no ground validation, so, only average overal merit.
Authors’ Responses:
Thanks to the comments of the reviewers. Due to the remoteness and hard-accessibility of the NW Yakutia, as mentioned in the Introduction, yet the area is key to many aspects of geocryology, this paper has preliminarily achieved coarse resolution IWP ground surface deformation monitoring and related factor change monitoring. Due to the lack of field data and other existing problems, we would further and progressively improve these aspects in the future research.

Round 2
Author Response
Dear Editors and Reviewers,
We thank the Remote Sensing editors and reviewers for their kind and valuable comments on this article. After carefully checking the expert opinions, we have revised the paper point-to-point according to the expert opinions and have answered all the raised opinions. The revised manuscript is marked under each question in red. The following are the expert opinions we revised and answered point-to-point.
Reviewer 2:
Review for: “Monitoring ground surface deformation of ice wedge polygon areas in Saskylakh, NW Yakutia, using Interferometric Synthetic Aperture Radar (InSAR) and Google Earth Engine (GEE)”
06-02-2023
Dear authors, dear editor,
I was happy to see that the authors have made a substantial amount of changes in a short timeframe, which has in my view improved the paper. Particularly the structure of the paper and the justification of the study design were improved. Below I will list additional comments regarding the revised text and the way previous comments were addressed. I hope the authors will be able to use this to improve the manuscript, which in my view still needs significant improvement in order to be publishable.
Main comments:
- The authors now give some hands-on recommendations for future work, which is useful. However, the implications of the results are still hardly discussed. I see that the authors have added a general description of increases in temperatures, ndvi and permafrost thaw, and a few sentences on anthropogenic disturbances but is rather unclear how this relates to the rest of the manuscript. In the discussion the authors should evaluate how your findings relate to the existing body of literature (e.g. are the changes fast or slow compared to other regions?), and what new insights the study provide (e.g. can your results give an indication about the degree of ground subsidence we should expect in contrasting thermokarst landforms and under different vegetation and moisture conditions in a warming Arctic?). Will LCPs or HCP, or wet or dry sites show stronger subsidence, for instance? If these “lessons learned” would be included in the discussion and at the end of the abstract, the study would be much more meaningful and clearer for a broad range of readers.
Authors’ Responses:
Thank you very much for your suggestions on the discussion section, we accepted your suggestions and made changes, which are very important to improve the quality of the article. The specific modifications are as follows:
We summarized the characteristics of the permafrost in Saskylakh and discussed each of them in comparison with other regions of the same latitude, also focusing on the surface deformation by human engineering.
Revised manuscript:
Lines 435-501:
5.1 Changes in the permafrost environment
In recent years, ground temperatures and LST have been rising in the Arctic and the hydroclimate variability is on the rise. Higher ground temperatures and LST have led to greater thaw depths, larger ALT, and lusher vegetation [62]. Permafrost in Saskylakh shows the characteristics of low LST and small ALT, but the vegetation coverage increases and the arctic greening is obvious. According to the prediction of Pearson et al [62] the vegetation types in Siberia, Alaska, and other regions in the Arctic will change, and the range of shrubs and trees has expanded and will expand sharply. This paper also found that the characteristics of vegetation coverage increase in Saskylakh, including taller, denser, and lusher vegetation in the IWP areas, with rapidly and extensively emerging bushes and trees. In this study area at Saskylakh, NDVI has shown a fluctuating upward trend in the last 20 years (Figure 6); both NDVI and/or vegetation growth rates are greater than those on the Arctic Coastal Plain of northern Alaska at the same latitude [63]. Com-pared with the LST in areas at the same latitude, the variation range of LST of Saskylakh is smaller than that of Northern Québec, Nunavik, Canada, and of the North Slope of northern Alaska, USA [56]. Globally, ALT and its average annual rate of change in Saskylakh are relatively small. For example, ALT in Mongolia and on the Qinghai-Tibet Plateau is generally found between 100 and 300 cm, and the ALT in Nordic countries and the Alps even exceeds 300 cm [29].
In recent years, increasing human activities have affected the Arctic eco-environment. In particular, the destruction or modifications of vegetation and the reduction of plant species diversity by infrastructure construction and operation also have further aggravated the permafrost degradation in the Arctic and Boreal zones [64,65]. In this study, the Saskylakh Aeroport under engineering impacts or other anthropogenic dis-turbances has severe GSD. A large range of settlement funnels are formed in the tower zone and a banded surface uplift is formed along the Aeroport runway. The elevational difference in the study area is less than 10 m. The major influences of elevation on the IWPs and the permafrost environment are reflected by the formation and expansion of thermokarst lakes and ponds on low-lying terrains. The latter is one of the important signals of IWP degeneration and permafrost degradation.
Studies have demonstrated that thermokarst activation and expansion are important indicators of IWP degeneration [15,66,67]. The areal extent of thermokarst lakes and ponds of Saskylakh shows a fluctuant change of increasing and then decreasing. A similar pattern was observed in a small area on the northeastern Qinghai-Tibet Plateau (QTP) but with a general shrinking trend of 41% for the period 1986-2015 [68]. Opposite on the central QTP was reported a thermokarst expansion of 123% in 50 years [69], while in northeastern Siberia, Kolyma lowland tundra, of only 4.5 % [70]. Because ice wedges are typically developed near the top of attached permafrost, polygonal terrain is particularly susceptible to thermokarsting. Thermokarst ponding affects the permafrost terrains when the thawing of ice-rich permafrost and the ensued meltwater drainage and enhanced soil erosion causes the loss of structural integrity of IWPs and the subsequent collapse of the ground [11,73-75].
5.2 GSD of ice wedge polygons (IWPs) and permafrost dynamics
The GSD of IWP areas is responsive to climate change [74]. The typical IWP regions and two transects show that in our study areas, with lowering ground temperature, the ground surface upheaved, and; with rising ground temperature, subsided (Figures 5, 8, and 9). This is consistent with earlier findings in the Arctic. For example, after a decade of automated multi-sensor monitoring of LST in the Adventdalen Valley, central Svalbard, Norway, Matsuoka et al. [8] found that ground upheaving mainly occurred upon the ground freezing generally started in late September to early November, and; ground surface subsidence, upon the ground thawing, generally starting in late May or early June.
The GSD in this paper is similar to the GSD of IWP areas monitored by Iijima et al. [75]. both of theirs and ours have achieved a wide range of monitoring, and cannot achieve a accuracy of a single IWP research. Ground-based measurements allow for fine-grained studies. For example, after a multi-year (1996-2018) leveling study in the Mackenzie delta area, Canada, Burn et al. [76] found the more intense settlement of ice-wedge troughs in comparison with that of IWPs. Similarly, we found that the deformation of ice-wedge troughs in the N - S transect (Figure 8) is larger than that in the surrounding areas. In addition, they found higher rates of GSD above ice wedges on hillslopes than those in other terrains because of the enhanced ground warming from downslope snow and wedge-ice meltwater runoff. It is worth noting that the GSD of Saskylakh Aeroport over arctic permafrost is more severe than that of natural ground surface under engineering disturbances. For example, the Saskylakh Airport has a large GSD; a settlement funnel has formed under and around the tower, and; the GSD on the airport runway has a rising trend. The GSD of the Iqaluit Airport on the Baffin Island, Canadian Arctic also has similar characteristics, and the GSD of the airport runway is larger than that of the surrounding area [77].
- The language will still need improvement in terms of grammar. In particular, the newly added paragraphs contain numerous grammatical inconsistencies, too many for me to list. Please seek advice from a native speaker, a professional in academic English or the journal itself. Several examples (but there are many more instances throughout the manuscript):
- Line 40-41: “Our study has preliminarily explored ground surface changes in IWP areas in Saskylakh and have provided referential information” should be: “Our study has preliminarily explored ground surface changes in IWP areas in Saskylakh and has provided referential information”.
- Line 27: “its” should be “their”.
- Line 42: change “have discussed” into “has discussed”.
- Line 95 – 101: Change “come from” to “come” and “based” to “based on”.
- Line 470: Change “LST is a low in thermokarst lakes” to “LST is low in thermokarst lakes”.
- Line 481: “is increasingly more affected by climate warming trend” should be “is increasingly affected by climate warming”.
Authors’ Responses:
We double-checked the paper for minor grammatical problems and made changes.
The above content proposed by the reviewer has been modified, and other modified parts are marked in red in the text.
- The results in line 360 to 370 (different dynamics of dry/wet, sparsely/lushly vegetated and low/high centre polygons) seem to provide too little basis for the conclusion you draw; the variability among IWPs of the same type seems to outweigh the differences among the types, especially with so few data points. You argue that you selected representative end members, which I can understand. But there is still not way of telling how representative the results from these endmembers are for the tundra area you study, based on the data you provide. For statistically meaningful comparisons, you would need at least 5 replicates for each unique combination of low/high centre, sparse/lush vegetation or wet/dry. Otherwise, the evidence should be treated as anecdotal. And from the changes in the abstract, I do not see this more “descriptive” nature come forward much.
Authors’ Responses:
We have removed that section, modified section is 4.2 GSD rate in the study area and different areas of ice wedge polygons (IWPs).
Minor comments:
- Line 30-31: January 2018 should be June 2018 I suppose? At least that is the timeframe reported in the methods and results.
Authors’ Responses:
Thank you for the reminder, we changed January 2018 to June 2018 in Line.
Revised manuscript:
Lines 29-31:
The results show an annual ground surface deformation rate (AGSDR) in Saskylakh at -49.73 to 45.97 mm/a during the period from 1 June 2018 to 3 May 2019.
- Line 32: I would suggest saying that “Several (n = 8) representative points” were selected, so that it is clear that the authors manually select endmembers.
Authors’ Responses:
We rewrote the section and we selected 5 Aeroport, 6 LCP and 6 HCP areas.
Revised manuscript:
Lines 36-37:
The GSDs are negatively correlated with the LST of most (low-centered polygons) (LCP) LCPs and (high-centered polygons (HCP) HCPs.
- Line 38: change “due to” to “reflected in”.
Authors’ Responses:
We changed “due to” to “reflected in” in Lines 37-38.
Revised manuscript:
Lines 38-41:
An evident permafrost degradation has been observed in Saskylakh as reflected in higher ground temperatures, lusher vegetation, greater active layer thickness, and fluctuant numbers and areal extents of thermokarst lakes and ponds.
- Referring back to my original minor comment nr. 2, also in this updated version the abstract only states that “information has been provided” and “processes have been discussed”. It does not tell the reader what exactly this information is, or what was observed and concluded. Please briefly but explicitly state what new insights the study has generated, instead of the current additions.
Authors’ Responses:
After discussion, we felt that this section contributed less to the abstract, and we removed it.
- Line 86-87: “man-made surfaces are in good contrast with different types of IWP”. Please clarify (in the manuscript itself) what you mean by “being in good contrast”? Perhaps you mean that this environment contains both natural and anthropogenic landforms?
Authors’ Responses:
“man-made surfaces” in here is the Saskylakh Aeroport. Here we would like to analyze the differences in surface deformation and other environmental factors between the artificially constructed area and the natural ground.
- Line 88-89 and line 140-142 seem to indicate that you want to perform a comparative study, that compares the degree of anthropogenic disturbance across man-made and natural environments? But the results and discussion do not show this at all. It seems better to omit this here since the results and discussion mainly deal with different types of ice wedge polygons and relations between several derived datasets.
Authors’ Responses:
We want to compares the degree of anthropogenic disturbance across man-made and natural environments, and we added discussion about it in subsection 5.2 GSD of ice wedge polygons (IWPs) and permafrost dynamics.
Revised manuscript:
Lines 497-503:
It is worth noting that the GSD of Saskylakh Aeroport over arctic permafrost is more severe than that of natural ground surface under engineering disturbances. For example, the Saskylakh Airport has a large GSD; a settlement funnel has formed under and around the tower, and; the GSD on the airport runway has a rising trend. The GSD of the Iqaluit Airport on the Baffin Island, Canadian Arctic also has similar characteristics, and the GSD of the airport runway is larger than that of the surrounding area [77].
- Table 2: For the year-averaged data (NDVI, MNDWI, LST), which time period was included? How many years of data did that result in? And which months or which time period did you include within the different years (whole year or only summer)? Please mention and explain your choices in the text.
Authors’ Responses:
Landsat 7 and 8 were used to extract NDVI, MNDWI, LST in 2010-2020, not every year's data meets the filtering requirements, and the filtered data are largely concentrated in the summer.
Revised manuscript:
Line 167:
Table 2. Optical and DEM data parameters used in this paper
Sensor |
Spatial resolution |
Parameter |
Bands |
Unit |
Range |
Time |
Value type |
Landsat 8 OLI/TIRS |
30 m |
LST |
B4, B5, B10 |
℃ |
/ |
2015~2020 |
Year-Average |
NDVI |
B4, B5 |
/ |
-1 ~ 1 |
Year-Average |
|||
MNDWI |
B3, B6 |
/ |
-1 ~ 1 |
Year-Average |
|||
Landsat 7 ETM+ |
NDVI |
B3, B4 |
/ |
-1 ~ 1 |
2010-2015 |
Year-Average |
|
MNDWI |
B2, B5 |
/ |
-1 ~ 1 |
Year-Average |
|||
MODIS (MOD11A1) |
1,000 m |
LST |
/ |
℃ |
/ |
2000-2020 |
Month- Average |
ALOS |
30 m |
DEM |
/ |
m |
/ |
/ |
/ |
- Line 152: “Representative examples of different IWPs. Insets (1) to (4) are the representative sample from Google earth images of HCP, Sparse vegetation LCP (SV-LCP), HCP Lush vegetation LCP (LV-LCP), Sparse vegetation HCP (SV-HCP), and Lush vegetation HCP (LV- HCP) in the study area, and; Insets (d5) and (d6) are the representative samples of Google Earth images for the Saskylakh Aeroport and tundra, respectively.” I think this description contains a few “HCP”s and “LCP”s too many? For instance: do you mean “Sparse vegetation LCP (SV-LCP)” instead of “HCP Sparse vegetation LCP (SV-LCP)” in line 153?
Authors’ Responses:
We have rewritten the section to no longer classify LCP and HCP, but to directly analyze the deformation characteristics of LCP and HCP.
Revised manuscript:
Lines 276-292:
3.6.4 Selection of ice wedge polygon (IWP) areas
GSD in permafrost areas is strongly related to variations in LST and ground ice con-tent [58-60]. In this paper, the GSD characteristics were compared with MODIS-based LST data. By calculating the PCC (GSD&LST) between the GSD of all ground surface deformation points in the study area and the regional average LST of the study area, the distribution of PCC (GSD&LST) in AGSDR reflects the overall GSD characteristics of the study area. Further, to analyze the GSD characteristics of different IWPs and Aeroport, five Aeroport, six LCP, and six HCP regions were randomly selected by circles with a radius of 50 m for each re-gion to have a similar number of deformation points in each region (Figure 2). To reduce the effect of individual deformation point on the GSD, we calculated the average GSD within each circle to represent each region. These GSD were then plotted separately as Aeroport, LCP, and HCP categories and analyzed in relation to LST.
Figure 2. Locations of different types of ice wedge polygons (IWPs) and N-S and W-E transects in Saskylakh, northwestern Yakutia, Siberian Arctic, Russia. Notes:
LCP: Low-centered polygon,
HCP: High-centered polygon
Revised manuscript:
Lines 336-355:
In general, the PCC(GSD&LST) is larger for points with larger AGSDR, and the PCC (GSD&LST) uncertainty is larger for AGSDR in -10 ~10 mm/a. The PCC(GSD&LST) is positive for points with negative AGSDR and negative for points with positive AGSDR (Figure 5a).
Further, the GSD characteristics of Aeroport, LCP, and HCP are also different. The GSD of the five Aeroport regions is larger than that of the LCP and HCP regions. Aeroport 1 is located in the tower of Aeroport, and the GSD shows a significant subsidence trend and is positively correlated with LST. Aeroport 2 to Aeroport 5 regions are located in the Aeroport runway, and; the GSD shows different degrees of uplift and is negatively corre-lated with LST (Figure 5b). The GSDs of most LCPs and HCPs were negatively correlated with LST, and only the GSD of region HCP3 was weakly positively correlated with LST. The GSD of HCP was greater than the GSD of LCP (Figure 5b).
Figure 5. Ground surface deformation (GSD) of different IWP sites in representative areas in Saskylakh, northwestern Yakutia. (a)is the distribution of correlation between GSD and LST in AGSDR; (b) is the GST of LCP, HCP and Aeroport in different regions.
Notes:
LST: Land surface temperature calculated by MODIS,
LCP: Low-centered polygon,
HCP: High-centered polygon,
PCC: Pearson correlation coefficient.
- Similar, for the month-averaged data: which months (only summer months or whole year?) were considered, and how many years? And why are there no month-specific data anywhere in the results, but only year-averages? Please make sure the data selection is clear and reproducible and that the table matches what is shown in the results.
Authors’ Responses:
Because the ground surface deformation data are monthly data, in order to study the relationship between deformation data and LST, monthly averaged LST is needed, but Landsat data can only satisfy annual averaging, not monthly averaging, MODIS data are used, and the relationship between deformation and temperature is studied using MODSI monthly averaged data.
Revised manuscript:
Line 167:
Table 2. Optical and DEM data parameters used in this paper
Sensor |
Spatial resolution |
Parameter |
Bands |
Unit |
Range |
Time |
Value type |
Landsat 8 OLI/TIRS |
30 m |
LST |
B4, B5, B10 |
℃ |
/ |
2015~2020 |
Year-Average |
NDVI |
B4, B5 |
/ |
-1 ~ 1 |
Year-Average |
|||
MNDWI |
B3, B6 |
/ |
-1 ~ 1 |
Year-Average |
|||
Landsat 7 ETM+ |
NDVI |
B3, B4 |
/ |
-1 ~ 1 |
2010-2015 |
Year-Average |
|
MNDWI |
B2, B5 |
/ |
-1 ~ 1 |
Year-Average |
|||
MODIS (MOD11A1) |
1,000 m |
LST |
/ |
℃ |
/ |
2000-2020 |
Month- Average |
ALOS |
30 m |
DEM |
/ |
m |
/ |
/ |
/ |
- If the Landsat LST is not used anymore (see lines 288-289) then remove this dataset from table 1 and line 285-286.
Authors’ Responses:
Landsat based LST were used in 4.5 Permafrost environment reflected by transects.
- Line 193-198: This sounds like a wonderful dataset if it’s so thoroughly compared to field data! But the cited paper (Beer et al., 2013) does not seem to refer to the same dataset you used (the Permafrost CCI dataset)? Please provide the appropriate reference.
Authors’ Responses:
We selected appropriate references and compared them. We compared the ALT of circumpolar active layer monitoring (CALM) and found little difference between the two, with the ALT of Saskylakh being around 1 m.
Revised manuscript:
Lines 180-192:
The ALT provided by permafrost CCI 2.3 is highly consistent with ALT provided by the Circumpolar Active Layer Monitoring (CALM) program [29].
- Luo, D.; Wu, Q.; Jin, H.; Marchenko, S. S.; Lü, L.; Gao, S. Recent changes in the active layer thickness across the northern hemisphere. Environ Earth Sci. 2016, 75(7), 555. doi: 10.1007/s12665-015-5229-2
- Line 183: The link to the CCI permafrost data no longer works, please update it.
Authors’ Responses:
We updated the link of CCI permafrost data.
Revised manuscript:
Lines 166-167:
Permafrost data used in this paper were provided by the ESA Permafrost Climate Change Initiative (CCI) 2.3 program (URL: https://climate.esa.int/en/odp/#/dashboard).
- Line 307: please indicate how you compared it (pearson’s correlation coefficient), and what your significance criterium was (e.g. alpha = 0.05).
Authors’ Responses:
We modified the description of this section according to Q23 and removed the correlation calculation.
- The addition in lines 272-274 is not really necessary I think, I merely wanted you to mention the threshold or other criterium for labelling something as a “pond” or not, so that others could reproduce the workflow.
Authors’ Responses:
The addition of this section is useful to illustrate the identification and monitoring of A thermokarst ponds or lakes, and we kept it. We added the threshold used in thermokarst ponds or lakes identified.
Revised manuscript:
Lines 261-262:
The area with MNDWI greater than zero is thus defined as thermokarst ponds or lakes.
- Line 324: Can you explain in the text why you chose 30m sampling distance, even though the spatial resolution of many of the layers is much coarser (e.g. 1km or more). Please indicate how you accounted for this spatial pseudoreplication.
Authors’ Responses:
30 m is the resolution of the most images, so we chose a sampling distance of 30 m. The sampling points were used to extract values of Altitude, AGSDR, MNDWI, NDVI, LST and ALT, the spatial resolution of Altitude, MNDWI, NDVI and LST (Landsat based) is 30 m, AGSDR is less than 30m, the resolution of ALT is 1000m.
- Line 365-366: The MNDWI of the IWP in subpanel c is similar to that of the dry IWPs (a,b,e), so why is this IWP not considered a dry one? Especially since it shows a very different response to LST that the other dry ones. So perhaps not all dry IWPs show a negative correlation with LST? Please either explain why subpanel c is not dry, or include it in the “dry” IWPs.
Authors’ Responses:
We have rewritten the section to no longer classify LCP and HCP, but to directly analyze the deformation characteristics of LCP and HCP.
Revised manuscript:
Lines 333-343:
In general, the PCC(GSD&LST) is larger for points with larger AGSDR, and the PCC (GSD&LST) uncertainty is larger for AGSDR in -10 ~10 mm/a. The PCC(GSD&LST) is positive for points with negative AGSDR and negative for points with positive AGSDR (Figure 5a).
Further, the GSD characteristics of Aeroport, LCP, and HCP are also different. The GSD of the five Aeroport regions is larger than that of the LCP and HCP regions. Aeroport 1 is located in the tower of Aeroport, and the GSD shows a significant subsidence trend and is positively correlated with LST. Aeroport 2 to Aeroport 5 regions are located in the Aeroport runway, and; the GSD shows different degrees of uplift and is negatively corre-lated with LST (Figure 5b). The GSDs of most LCPs and HCPs were negatively correlated with LST, and only the GSD of region HCP3 was weakly positively correlated with LST. The GSD of HCP was greater than the GSD of LCP (Figure 5b).
Figure 5. Ground surface deformation (GSD) of different IWP sites in representative areas in Saskylakh, northwestern Yakutia. (a)is the distribution of correlation between GSD and LST in AGSDR; (b) is the GST of LCP, HCP and Aeroport in different regions.
Notes:
LST: Land surface temperature calculated by MODIS,
LCP: Low-centered polygon,
HCP: High-centered polygon,
PCC: Pearson correlation coefficient.
- In addition to above; should subpanel d be considered “wet”? It has the highest MNDWI of all.
Authors’ Responses:
Ibid
- Paragraph 4.3 and figure 6 (which is much improved, thank you!): please state more clearly in the text and figure labels that this is modelled alt and permafrost temperature data, not measured. Please add in the legend whether this is year-round or seasonal (see also comment 10). If these are yearly averages of NDVI, then how did you account for the confounding effects of snow cover duration and difference in image availability among years?
Authors’ Responses:
We stated clearly the ALT and permafrost temperature data used is modelled ALT and permafrost temperature data.
Landsat 7 and 8 data in summer satisfied the filter condition and was explained in Lines 157-159.
Revised manuscript:
Lines 356-357:
The modeled average ground temperatures and ALT in 1997-2020 were selected for analysis.
Lines 157-159:
A filter of 10% cloud cover has been used and only images of Landsat 7 ETM+ and 8 OLI/TIRS during summer met this requirement.
- Line 419-424: Thank you for this addition based on my previous comment. I am afraid I was unclear; I wanted to see some discussion or evidence whether the trend of thermokarst area / ponds over time was significant (i.e., is the increase in area over time statistically significant, and is the increase in number over time significant?). Because you report that the area was 1.36 km2 in 2000 and 1.62 km2 in 2020 and call this an “increase”, but figure 7 shows large differences over time, and something that looks more like high fluctuations and a increase followed by a decrease. If you want to call this an increase, please provide an appropriate statistic (ideally mann kendall or theil sen trend), or don’t call it an increase.
Authors’ Responses:
We accepted your advice and changed the increase into fluctuations.
Revised manuscript:
Lines 376-378:
The areal extent of thermokarst ponds fluctuated from 2000 to 2020 (Figures 7a and 7b). The areal extent of thermokarst ponds was first expanding (2000-2010) and then shrinking (2010-2020).
- Paragraph 4.5: This paragraph does not contain results, only description of the approach. Please merge it with paragraph 3.6
Authors’ Responses:
We consider this section to be the result of the reflection of transects and therefore did not merge it.
Revised manuscript:
Lines 426-432:
It is found from the transects that: (1) There are only small elevation differences in the study area; (2) Thermokarst lakes and ponds are mainly found on the low-lying terrains; (3) LST is lower in thermokarst lakes and ponds; (4) In the areas of thermokarst lakes and ponds, no deformation results have been yielded due to the SAR data incoherence of water surfaces; (5) The active layer gradually thins northwards from 66.5 cm in the south to 56 cm in the north; (6) The Saskylakh Aeroport is a typical representative of engineering dis-turbance, with high LST and low NDVI, and possibly large ALT.
- Figures 8 & 9: In line 288-289 you state that you used modis lst, not landsat lst. Yet, the spatial resolution of the lst data in figure 8 and 9 seem to indicate 30m intervals, not 1000m intervals. Please explain why this is the case and adjust in the manuscript. I think you need to adjust the methods (line 288-289) to explain that here you probably did use Landsat LST.
Authors’ Responses:
We explained the data used in Figures 8 and 9 in subsection 3.6.5 Transect selection.
Revised manuscript:
Lines 291-293:
Two perpendicular transects, across the study area were selected to analyze the spatial heterogeneity of GSD and the responses of IWPs to changes in environmental factors (elevation, Landsat based NDVI, MNDWI, LST, and MODIS based ALT) (Figure 2).
- Line 488-489: “The elevational difference in the study area is less than 10 m, which has little influence on IWPs and permafrost”. How do you conclude this? The data would only show a correlation, not an influence, and no correlations or other evidence are provided. Please provide evidence for this statement or alter it (e.g. “altitude shows little association with”)
Authors’ Responses:
Accepted and modified. Our previous narrative was inaccurate and we have removed it
Revised manuscript:
Lines 459-463:
The elevational difference in the study area is less than 10 m. The major influences of elevation on the IWPs and the permafrost environment are reflected by the formation and expansion of thermokarst lakes and ponds on low-lying terrains. The latter is one of the important signals of IWP degeneration and permafrost degradation.
- Figure 10: please remove p values from this figure. The significance will depend heavily on the sampling distance along the transect and is subject to spatial pseudoreplication (i.e. spatially correlated residuals are not accounted for and the same permafrost cci pixel will be sampled multiple times). In addition, NDWI and NDVI are always significantly correlated since they use similar bands. Including lakes in this analysis also inflates p-values since lakes will differ strongly in all surface properties per definition. Hence, the use of p values is not warranted or even meaningful here.
Authors’ Responses:
Thank you for your advice.
According to Q30, we removed figure 10.
- Line 507 is a repetition of line 486. Please remove redundant information in this paragraph.
Authors’ Responses:
Accepted and modified. We removed redundant information in this paragraph.
- Line 513: How do meaningfully obtain a range for both types, with only 4 examples of both types? Please explain in the text.
Authors’ Responses:
We implemented different experimental approaches and found that LCP and HCP were negatively correlated with LST.
- I don’t think you can derive this from figure 10, because it shows lst data that are averaged over many years and seasons, against gsd of a single year. So that does not support conclusions about relations between changes in temperature and subsidence.
Authors’ Responses:
Accepted and modified. We removed figure 10.
- Line 539-540: But you have monitored gsd in several distinct iwps in figure 5 right? Perhaps I misunderstand this sentence, can you clarify it? (also “temprarily” -> temporarily). Do you mean you cannot monitor them in the field right now?
Authors’ Responses:
We implemented different experimental approaches and found that LCP and HCP were negatively correlated with LST
Revised manuscript:
Lines 341-343:
The GSDs of most LCPs and HCPs were negatively correlated with LST, and only the GSD of region HCP3 was weakly positively correlated with LST. The GSD of HCP was greater than the GSD of LCP (Figure 5b).
- Line 548-550: please describe what you found and what it means, rather than repeating what you have studied.
Authors’ Responses:
We rewrote it and described the findings. A comparison with existing studies was made. The ground surface deformation characteristics of airports in permafrost areas were summarized.
Revised manuscript:
Lines 485-500:
The GSD in this paper is similar to the GSD of IWP areas monitored by Iijima et al. [75]. both of theirs and ours have achieved a wide range of monitoring, and cannot achieve a accuracy of a single IWP research. Ground-based measurements allow for fine-grained studies. For example, after a multi-year (1996-2018) leveling study in the Mackenzie delta area, Canada, Burn et al. [76] found the more intense settlement of ice-wedge troughs in comparison with that of IWPs. Similarly, we found that the deformation of ice-wedge troughs in the N - S transect (Figure 8) is larger than that in the surrounding areas. In addition, they found higher rates of GSD above ice wedges on hillslopes than those in other terrains because of the enhanced ground warming from downslope snow and wedge-ice meltwater runoff. It is worth noting that the GSD of Saskylakh Aeroport over arctic permafrost is more severe than that of natural ground surface under engineering disturbances. For example, the Saskylakh Airport has a large GSD; a settlement funnel has formed under and around the tower, and; the GSD on the airport runway has a rising trend. The GSD of the Iqaluit Airport on the Baffin Island, Canadian Arctic also has similar characteristics, and the GSD of the airport runway is larger than that of the surrounding area [77].
- Conclusion: I am still sceptical about conclusions 2 and 3 since they are based on so few examples per case, and it seems (based on MNDWI in figure 5) that not all dry and wet IWP showed the same behaviour relative to LST. Please tone down the conclusions (e.g. “a subset of several LCP and HCP suggests that …”)
Authors’ Responses:
We wrote our conclusions based on the new experimental results.
Revised manuscript:
Lines 508-519:
1) The AGSDR in Saskylakh ranged from 49.73 to 45.97 mm/a in 2018-2019, and in 75% of the study area, the AGSDR was between 10 ~ 10 mm/a.
2) All GSD points in the region indicated a closer relationship between GSD and LST for our observational points with larger AGSDR, and positive correlation between GSD and LST for those points with negative AGSDR, and; vice versa.
3) Further, the GSDs of the five Aeroport areas indicated the most drastic defor-mation under engineering disturbances in the Saskylakh Aeroport area, and; the GSDs of the six LCP and six HCP areas indicated that the GSDs of most LCPs and HCPs are negatively correlated with LST.
4) Arctic permafrost in Saskylakh showed a trend of degradation, as evidenced by rising ground temperature and NDVI (Shown as denser and taller vegetation), deepening active layer, and expanding thermokarst lakes and ponds.
- Line 569-570: Please explain in the text what you mean by the unified resolution of the datasets. (Otherwise, the information added in line 568-587 is very useful and to the point!)
Authors’ Responses:
Because the resolution difference between Landsat and MODIS is too large. The unified resolution is similar resolution, which has been modified in the article.
Revised manuscript:
Lines 521-523:
There are some inadequacies in this study, such as the limited penetration ability of the C-band Sentinel-1 satellite, the inconsistent resolution of optical satellites, the similar spatial resolution of data sources, and the lack of ground-based measurement support.
